# Beyond Communication Overhead: A Multilevel Monte Carlo Approach for Mitigating Compression Bias in Distributed Learning

Ze'ev Zukerman [* 1]   Bassel Hamoud [* 1]   Kfir Y. Levy [1]

## Abstract

Distributed learning methods have gained substantial momentum in recent years, with communication overhead often emerging as a critical bottleneck. Gradient compression techniques alleviate communication costs but involve an inherent trade-off between the empirical efficiency of biased compressors and the theoretical guarantees of unbiased compressors. In this work, we introduce a novel Multilevel Monte Carlo (MLMC) compression scheme that leverages biased compressors to construct statistically unbiased estimates. This approach effectively bridges the gap between biased and unbiased methods, combining the strengths of both. To showcase the versatility of our method, we apply it to popular compressors, like Top-$k$ and bit-wise compressors, resulting in enhanced variants. Furthermore, we derive an adaptive version of our approach to further improve its performance. We validate our method empirically on distributed deep learning tasks.

## 1. Introduction

Distributed learning has emerged as a critical paradigm for scaling machine learning to massive datasets across multiple computing nodes. In this setting, a central server coordinates multiple worker nodes, each computing local gradients on their respective data shards and communicating updates back to the server. This parallelization accelerates training, but introduces a fundamental bottleneck: communication overhead (Konečný et al., 2018; Wang et al., 2021). To mitigate this, gradient compression techniques are commonly employed to reduce the volume of transmitted data (Alistarh et al., 2017; Lin et al., 2018). However, these methods introduce a trade-off between unbiased and biased compressors (Beznosikov et al., 2020).

Unbiased compression techniques, such as random sparsification (e.g., Rand-$k$) and statistical quantization methods (e.g., QSGD (Alistarh et al., 2017)), ensure that the expected value of the compressed gradient remains equal to the original gradient. They are well understood theoretically because they align with the standard theoretical guarantees of data-parallel SGD (Jain et al., 2017; Dekel et al., 2012). However, their empirical performance is often suboptimal since they select elements at random rather than prioritizing the most informative components of the gradient. This leads to inefficient gradient updates, which negatively affect performance.

In contrast, biased compressors, such as Top-k sparsification, retain the most informative components of the gradient while discarding less significant elements, leading to superior empirical performance (Seide et al., 2014; Richtárik et al., 2021). However, they introduce a degradation in theoretical guarantees, as their biased nature prevents them from directly aligning with the classical analysis of data-parallel SGD. This necessitates additional correction mechanisms, such as error feedback (e.g., EF21 (Richtárik et al., 2021)), to ensure convergence.

Beyond gradient compression, distributed learning encompasses a wide range of techniques aimed at improving scalability and efficiency. Methods such as asynchronous training (Recht et al., 2011; Dean et al., 2012; Tyurin et al., 2024), in which worker nodes update the central model without waiting for all nodes to synchronize, help mitigate communication delays. Federated learning (Konecný et al., 2016; Kairouz et al., 2021), which enables training while preserving data privacy, has also gained significant traction. Furthermore, decentralized training (Koloskova et al., 2019) removes the need for a server to maintain the model and instead propagates knowledge through "gossip" mechanisms. Additionally, techniques like local updates (Stich, 2019; Dahan & Levy, 2024; Mishchenko et al., 2022; Condat et al., 2023), where workers perform multiple gradient steps before communicating with the server, reduce communication frequency and enhance efficiency. Each of these methods

---
*Equal contribution   [1]Viterby Faculty of Electrical and Computer Engineering, Technion, Israel. Correspondence to: Ze'ev Zukerman <ze.zukerman@campus.technion.ac.il>, Bassel Hamoud <bassel164@campus.technion.ac.il>, Kfir Y. Levy <kfirylevy@technion.ac.il>.

*Proceedings of the 42^{nd} International Conference on Machine Learning*, Vancouver, Canada. PMLR 267, 2025. Copyright 2025 by the author(s).

aims to strike a balance between computation, communication, and convergence guarantees.

To bridge the gap between unbiased and biased compression techniques, we introduce a novel compression scheme based on Multilevel Monte Carlo (MLMC) methods (Giles, 2013). MLMC techniques construct an estimator by combining multiple levels of approximation, each with a different quality (variance) and cost. The heart of the MLMC method is that it transduces bias into variance. We leverage this core property to construct unbiased estimates from biased compressed gradients, transducing their bias into controlled variance, and thereby ensuring both empirical efficiency and good parallelization ability.

We apply our MLMC-based framework to popular compressors, demonstrating how it enhances their performance by reducing compression bias while keeping the communication costs minimal. Furthermore, we introduce an adaptive version of our approach, dynamically optimizing compression levels to further improve efficiency. We validate our method through various deep learning experiments, showcasing its convergence speed and communication efficiency.

By leveraging MLMC to mitigate compression bias, our work provides a principled solution that reconciles the strengths of biased and unbiased compression techniques. This contribution paves the way for more efficient distributed learning frameworks that maintain both strong theoretical guarantees and superior empirical performance.

### 1.1. Related Work

Gradient compression techniques are essential for reducing communication costs in distributed optimization. These methods fall into *unbiased* and *biased* approaches, each offering different trade-offs in terms of convergence guarantees and empirical performance. Some works also explore *bidirectional compression*, where both worker-to-server and server-to-worker communication is compressed (Horváth et al., 2022; Gorbunov et al., 2020). While bidirectional compression is relevant in some distributed learning settings, our focus remains on *gradient compression*, where the primary challenge is reducing worker-to-server communication while ensuring convergence.

**Unbiased Compression Methods.** Unbiased compression methods ensure that the expectation of the compressed gradient equals the true gradient. QSGD (Alistarh et al., 2017) and natural compression (Horváth et al., 2022) are prominent examples, providing strong theoretical guarantees but suffering from slow empirical convergence due to the random selection of gradient components. DIANA (Mishchenko et al., 2023; Horváth et al., 2019) overcomes this by compressing gradient differences. MARINA (Gorbunov et al., 2021) incorporates variance reduction to miti-

gate this issue by using unbiased compressions of gradient differences. DASHA (Tyurin & Richtárik, 2023) improves efficiency using structured and compressed updates only. EF-BV (Condat et al., 2022) offers a unifying framework for biased and unbiased compressors, which recovers both DIANA and EF21 (Richtárik et al., 2021) as special cases, but does not aim to generate unbiased estimators from biased ones, in contrast to our work. Horváth & Richtárik (2021) developed a related approach, which constructs an unbiased compressor from two biased ones using an error feedback mechanism, achieving better convergence at the cost of roughly doubling the communication cost.

**Biased Compression Methods and Error Feedback.** Biased compressors, such as Top-K sparsification (Stich et al., 2018) and SignSGD (Bernstein et al., 2018; Karimireddy et al., 2019), retain the most informative gradient components, leading to superior empirical performance. However, these methods introduce biases that require correction to ensure convergence. Error feedback (EF) (Seide et al., 2014) was introduced as a correction mechanism, which was later refined by EF21 (Richtárik et al., 2021) to eliminate restrictive assumptions and improve theoretical guarantees. EF21-SGDM (Fatkhullin et al., 2023) further stabilizes updates using momentum, reducing sample complexity and improving convergence speed. Adaptive gradient sparsification (Han et al., 2020) dynamically adjusts sparsity levels, balancing communication efficiency and performance. The introduction of bias typically adds additional terms to the convergence bounds (Fatkhullin et al., 2023) which can hinder performance in massive parallelization settings.

In summary, unbiased methods are well understood, easy to analyze, and enjoy simple bounds, but are often impractical due to inefficiency, while biased methods, coupled with EF techniques, offer superior empirical results. Our work builds on these insights by further refining biased compression strategies and offering a plug-and-play mechanism to construct unbiased estimates from biased ones. We further show that our technique works seamlessly for any compressor and enhances convergence efficiency, bridging the gap between biased and unbiased compression methods.

## 2. Background

### 2.1. Problem Statement

We consider the distributed machine learning setting with a master server and $M$ machines $i = 1, ..., M$. We assume a heterogeneous setting in which each machine $i \in [M]$ has access to $i.i.d$ samples from some data distribution $\mathcal{D}_i$. We aim to minimize the following problem:

$$\arg \min_{x \in \mathbb{R}^d} f(x) = \arg \min_{x \in \mathbb{R}^d} \frac{1}{M} \sum_{i=1}^{M} f_i(x) \qquad (1)$$

where $f_i : \mathbb{R}^d \to \mathbb{R}$ measures the expected loss of the model on the local data of machine $i$. Namely, $f_i(x) = \mathbb{E}_{z_i \sim \mathcal{D}_i}[f_i(x, z_i)]$, where $f_i(x, z_i)$ is the loss of model $x$ w.r.t sample $z_i \sim \mathcal{D}_i$. In each step $t \in [T]$, the master server broadcasts the current model $x_t \in \mathbb{R}^d$ to the $M$ machines, and each machine $i \in [M]$ computes a stochastic gradient $v_{t,i} \triangleq \nabla f_i(x_t, z_{t,i})$, where $z_{t,i} \sim \mathcal{D}_i$, computes a compression (an estimate) $g_{t,i}$ of $v_{t,i}$, and sends $g_{t,i}$ back to the server. The server aggregates $\{g_{t,i}\}_{i=1}^M$ and uses the result to update the model. Note that when $g_{t,i} = v_{t,i}$, we have the known Data-parallel SGD scheme, which is formalized in Alg. 1 and Theorem 2.3 (Dekel et al., 2012; Ghadimi & Lan, 2013). Note that a central property of Theorem 2.3 is that the gradients are *conditionally unbiased*, i.e., $\mathbb{E}[v_{t,i}|x_t] = \nabla f_i(x_t), \forall t, i$. This assumption is not always satisfied when compression is introduced, as we elaborate in the following subsections.

We make the following assumptions throughout the paper:

**Assumption 2.1.** The loss functions $f_i$ are $L$-smooth: $f_i(y) \leq f_i(x) + \langle \nabla f_i(x), y - x \rangle + \frac{L}{2} \|y - x\|^2, \forall x, y \in \mathbb{R}^d, \forall i \in [M]$.

**Assumption 2.2.** The uncompressed stochastic gradients $\nabla f_i(x, z)$ have bounded variance: $\forall i \in [M], \forall x \in \mathbb{R}^d$, $\mathbb{E}[\|\nabla f_i(x, z) - \nabla f_i(x)\|^2 |x] \leq \sigma^2$.

---

**Algorithm 1** Data-parallel SGD

   **Input:** initialization $x_1$, step-size $\eta$.
   **for** $t = 1$ **to** $T$ **do**
      The server broadcasts $x_t$ to machines $i = 1, .., M$
      **for** $i = 1$ **to** $M$ **in parallel do**
         Sample $z_{t,i}$ from local dataset $\mathcal{D}_i$
         Compute $v_{t,i} = \nabla f_i(x_t, z_{t,i})$
         Send $v_{t,i}$ to server
      **end for**
      Server aggregates: $v_t = \frac{1}{M} \sum_{i=1}^M v_{t,i}$
      Server updates: $x_{t+1} = x_t - \eta v_t$
   **end for**

---

**Theorem 2.3.** *Under Assumption 2.1, Alg. 1 guarantees the following error in the convex case, for any $\eta \leq 1/2L$:*

$$\mathbb{E}[f(\bar{x}_T) - f(x^*)] \leq \frac{D^2}{2T\eta} + \frac{\eta}{T} \sum_{t=1}^T \mathbb{E}V_t^2,$$

*and the following error in the nonconvex case, for $\eta \leq 1/L$:*

$$\frac{1}{T} \sum_{t=1}^T \mathbb{E}\|\nabla f(x_t)\|^2 \leq \frac{2\Delta_1}{T\eta} + \frac{\eta L}{T} \sum_{t=1}^T \mathbb{E}V_t^2,$$

*where* $\bar{x}_T = \frac{1}{T} \sum_{t=1}^T x_t$, $x^* = \arg\min_x f(x)$, $D = \|x_1 - x^*\|$, $V_t^2 = \frac{1}{M^2} \sum_{i=1}^M \mathbb{E}[\|v_{t,i} - \nabla f_i(x_t)\|^2 |x_t]$, *and* $\Delta_1 = f(x_1) - f(x^*)$.

Note that the error bounds in Theorem 2.3 depend on the variance of the gradients. Furthermore, under assumption

2.2 and by optimizing over $\eta$, the error bound (both for the convex and the nonconvex cases) can be written as:

$$\mathcal{O}\left(\frac{1}{T} + \frac{\sigma}{\sqrt{MT}}\right) \tag{2}$$

Up to factors that are independent of $T, M$ and $\sigma$. As we elaborate next, incorporating compression into parallel SGD in Alg. 1 alters these bounds, either by increasing the variance term in the case of unbiased compression, or by rendering them obsolete in the case of biased compression.

### 2.2. Training with Compressed Gradients

While the naive parallelization scheme in Alg. 1 is straightforward, it neglects the communication cost between the machines and the server. With today's computational power, communication serves as the main bottleneck in the learning process. Consequently, many methods resort to using *compressed* versions of the gradients to reduce the communication cost (see Sec. 1.1). Such compressors can be broadly classified into two main categories:

(1) **Unbiased compressors**, which for $\omega \geq 0$ and $\forall x \in \mathbb{R}^d$ satisfy:

$$\mathbb{E}[C(x)] = x \; ; \; \mathbb{E}[\|C(x) - x\|^2] \leq \omega \|x\|^2, \tag{3}$$

(2) **Biased compressors**, which for $0 < \alpha \leq 1$ and $\forall x \in \mathbb{R}^d$ satisfy:

$$\mathbb{E}[C(x)] \neq x \; ; \; \mathbb{E}[\|C(x) - x\|^2] \leq (1 - \alpha) \|x\|^2 \tag{4}$$

where the above expectations are w.r.t. the randomization potentially introduced by $C$. The use of unbiased compressors is usually straightforward as they are easy to incorporate into the parallelization scheme in Alg. 1 where only the second term will be affected with an increased variance. However, biased compressors generally yield better practical results, since they tend to retain more energy of the compressed entity compared to unbiased counterparts. Unfortunately, their naive incorporation in Alg. 1 may fail to converge (Beznosikov et al., 2020), since now the compressed gradients are not unbiased estimates of the true gradients, and more sophisticated optimization schemes are required (Seide et al., 2014; Richtárik et al., 2021). We now survey a few popular compressors that are used for training with compressed gradients.

**Top-$k$** is a popular compressor which retains the largest $k$ elements in absolute value of a given vector and zeros the rest. Naturally, Top-$k$ is a biased compressor that satisfies Eq. (4) with $\alpha = k/d$ and $k$ between 1 and $d$. It is generally empirically superior to its prevalent unbiased counterpart, Rand-$k$, which retains $k$ randomly selected elements of a

vector. Moreover, we consider a generalization of Top-$k$, which we term *s-segmented Top-$k$*, or *s-Top-$k$*, which sorts a given vector of length $d$, divides it into segments of length $s$ (except perhaps the last segment), and retains the $k$ segments with the largest norm. Accordingly, $\alpha = sk/d$ and $k$ ranges from 1 to $\lceil d/s \rceil$. Note that regular Top-$k$ can be recovered from its generalized variant when $s = 1$.

**Bit-wise compressors** are methods that utilize binary representation to compress numerical data. In our setting, we perform bit-wise compression of the binary representation of each element in the gradient vector in an element-wise manner. There are two common approaches for bit-wise compression:

(1) **Fixed-point compressors.** Fixed-point methods encode numbers with fixed integer and fractional bits. Compression is done by discarding the least significant bits and keeping the $F$ most significant bits, introducing distortion that is bounded by $2^{-F}$ for each element.

(2) **Floating-point compressors.** Floating-point methods encode numbers using a *mantissa*, the fractional part, and an *exponent*, the scale factor. Floating-point compressors retain the exponent and the $F$ most significant bits of the mantissa, forming a biased compressor that satisfies Eq. (4) with $\alpha = 1 - 2^{-F}$.

As mentioned, the introduction of compression changes the error bounds of gradient-based methods. Unbiased compression only changes the variance. That is because $\mathbb{E}[C(v_{t,i})] = \mathbb{E}[\mathbb{E}[C(v_{t,i})]|x_t] = \mathbb{E}[v_{t,i}|x_t] = \nabla f_i(x_t)$, namely the compressed gradients are still unbiased estimates of the true gradient. Additionally, note that:

$$\mathbb{E}[\|C(v_{t,i}) - \nabla f_i(x_t)\|^2 | x_t]$$
$$= \mathbb{E}[\|C(v_{t,i}) - v_{t,i} + v_{t,i} - \nabla f_i(x_t)\|^2 | x_t]$$
$$= \underbrace{\mathbb{E}[\|C(v_{t,i}) - v_{t,i}\|^2 | x_t]}_{\sigma_{comp}^2} + \underbrace{\mathbb{E}[\|v_{t,i} - \nabla f_i(x_t)\|^2 | x_t]}_{\sigma^2},$$

and simply plugging this updated variance term into the known bounds of Theorem 2.3 yields the corresponding error bounds. The use of biased compressors, although empirically superior to their prevalent unbiased counterparts, requires a different treatment to account for the bias they introduce, which often hinders parallelization and affects the error bounds (Fatkhullin et al., 2023).

In our work, we suggest a novel approach to construct unbiased versions of popular compressors such that we retain the most important information (as biased compressors) without hindering parallelization (as unbiased compressors). We aim to construct new enhanced estimators that achieve the best of both worlds using a novel compression scheme based on Multilevel Monte Carlo on which we elaborate next.

## 2.3. Multilevel Monte Carlo methods

Monte Carlo methods construct a variance-reduced estimator for the expectation of some random variable $X$ using an ensemble of independent stochastic samples. In its simplest form, given *unbiased* i.i.d. samples $\{X^{(j)}\}_{j=1}^N$ of $X$, such that $\mathbb{E}[X^{(j)}] = \mathbb{E}[X], \forall j \in [N]$, a Monte Carlo estimate of $\mathbb{E}[X]$ is given by $\frac{1}{N} \sum_{j=1}^N X^{(j)}$. This estimator enjoys a reduced variance by a factor of $1/N$ compared to that of the individual samples. This method implicitly assumes that the cost and quality (variance) of each sample are identical.

Multilevel Monte Carlo (MLMC) methods (Giles, 2013) generalize this to a setting where we can access samples of increasing quality but at an increasing cost. MLMC methods also obviate the need for unbiased samples, unlike regular Monte Carlo. Namely, given samples $X^{l,(j)}$ with variance $V^l$ and cost $K^l$, for $j \in [N]$ and $l \in [L]$, where typically $V^l$ *decreases* while $K^l$ *increases* with $l$, the MLMC estimator of $\mathbb{E}[X]$ is given by:

$$\tilde{X} \triangleq X^0 + \frac{1}{p^l}(X^l - X^{l-1}), \quad \text{where} \quad l \sim p^l, \quad (5)$$

where $\{p_l\}_{l=1}^L$ is a non-zero probability distribution over the levels $l \in [L]$ and $X^l$ and $X^{l-1}$ are some estimators of $\mathbb{E}[X]$ based on samples of levels $l$ and $l - 1$, respectively. One of the most intriguing properties of the MLMC estimator is that it is a naturally *unbiased* estimator of the highest-level expectation, namely $\mathbb{E}[\tilde{X}] = \mathbb{E}[X^L]$. Furthermore, MLMC methods effectively transduce bias into variance. This important property will play a central role in our method, where $X^L$ will be an unbiased estimate of $\mathbb{E}[X]$, implying that $\tilde{X}$ is an unbiased estimate of $\mathbb{E}[X]$.

## 3. Multilevel Monte Carlo Parallel SGD

Motivated by the trade-off between biased compressors, which have superior performance but suffer worse theoretical guarantees, and unbiased compressors, which exhibit the opposite, we set to explore a method to bridge this gap. Namely, we pose the following question:

*Can we simultaneously utilize the superior performance of biased compressors and enjoy the better theoretical guarantees of unbiased compressors?*

To address this, we propose a novel method that exploits the properties of MLMC estimators and allows us to use biased compressors without adversely affecting the theoretical convergence guarantees. Our idea is to generate MLMC estimators of the biased gradient compressions and use those to update the model. This way, although the compressed gradients can be biased, their MLMC estimators are always unbiased, but are typically accompanied by a slightly increased variance.

Each compressor (e.g., Top-$k$, $s$-Top-$k$, bit-wise compressors, etc.) typically has a parameter that tunes the extent of compression. For example, a smaller $k$ in Top-$k$ or $s$-Top-$k$ translates to a more aggressive compression. We define the "estimate levels" $l \in [L]$ of the MLMC estimate in correlation with these parameters, such that lower levels correspond to more aggressive compression, while higher levels correspond to a softer compression. For efficiency, we incorporate this into a new class of compressors, which we term *Multilevel Compressors*, and define it as follows.

**Definition 3.1.** $C^l : \mathbb{R}^n \to \mathbb{R}$ is a multilevel compressor, where $l \in [L]$ corresponds to the compression level and the highest level $L$ corresponds to no compression, i.e., $\forall v \in \mathbb{R}^d : C^L(v) = v$.

For example, in the case of Top-$k$ and $s$-Top-$k$, the levels $l$ correspond to the parameter $k$. Thus, lower levels lead to a worse estimate of the original gradient but a lower communication cost, while higher levels yield a better estimate but a higher communication cost, and the highest level $L$ corresponds to no compression at all (e.g., Top-$k$ with $k = d$, or $s$-Top-$k$ with $k = \lceil d/s \rceil$).

Concretely, given a multilevel compressor $C^l$, where $l \in [L]$, non-zero level probabilities $\{p^l\}_{l=1}^L$, and an uncompressed stochastic gradient $v_{t,i}$, the MLMC gradient estimate of $v_{t,i}$ is given by:

$$\tilde{g}_{t,i} = g_{t,i}^0 + \frac{1}{p^l}(g_{t,i}^l - g_{t,i}^{l-1}), \quad \text{where} \quad l \sim p^l \quad (6)$$

where $g_{t,i}^l = C^l(v_{t,i}), g_{t,i}^{l-1} = C^{l-1}(v_{t,i})$, and we define $g_{t,i}^0 = 0$ (e.g., Top-$k$ with $k = 0$). This MLMC compression scheme yields an unbiased estimate of the true gradient in step $t$, $\nabla f_i(x_t)$, as we formalize in Lemma 3.2.

**Lemma 3.2.** *For any multilevel compressor $C^l, l \in [L]$ and any non-zero probabilities $\{p^l\}_{l=1}^L$, the MLMC estimator $\tilde{g}_{t,i} \triangleq g_{t,i}^0 + \frac{1}{p^l}(g_{t,i}^l - g_{t,i}^{l-1})$, where $l \sim p^l$, is a conditionally unbiased estimate of the true gradient, $\nabla f_i(x_t)$. Namely: $\mathbb{E}[\tilde{g}_{t,i}|x_t] = \nabla f_i(x_t), \forall t \in [T], \forall i \in [M]$.*

We defer the proof to App. A. Intuitively, our MLMC block can be thought of as a black box that takes the stochastic gradient, a compressor (e.g., $s$-Top-$k$), and a probability distribution over the compression levels (e.g., the values of $k$) and outputs an unbiased estimate of the true gradient. The probability distribution is optimized to minimize the variance of the MLMC estimator. In some cases, we show that the probability distribution can be chosen in an *adaptive* manner, per sample, to optimize the variance for each sample independently. Furthermore, since our method replaces the stochastic gradients with their MLMC estimates, which are also unbiased (see Lemma 3.2), the error bounds in Theorem 2.3, for the convex and the nonconvex case, remain largely the same and only the variance term is affected.

We formalize our method in Alg. 2, where each machine $i \in [M]$: (1) computes the gradient $v_{t,i}$ based on one stochastic sample $z_{t,i}$, (2) samples a compression level $l \in [L]$ according to a predefined probability distribution $\{p^l\}_{l=1}^L$, (3) constructs the MLMC gradient $\tilde{g}_{t,i}$ according to Eq. (6), and (4) sends it back to the server. The server aggregates the MLMC gradients and updates the model. Note that while the general template of Alg. 2 requires two compressions in each iteration for the levels $l$ and $l-1$, in certain cases computing the residual $g_{t,i}^l - g_{t,i}^{l-1}$ can be done efficiently without explicitly calculating each term, and it can be transmitted cheaply as well. For example, for Top-$k$, $g_{t,i}^l - g_{t,i}^{l-1}$ includes only the $l$'th largest element (in absolute value), and for $s$-Top-$k$, the residual includes the segment of length $s$ with the $l$'th largest norm.

---

**Algorithm 2** MLMC-Compressed Parallel SGD
___

**Input:** initialization $x_1$, step-size $\eta$, multilevel compressor $C^l$, and level probabilities $\{p^l\}_{l=1}^L$
**for** $t = 1$ **to** $T$ **do**
    The server broadcasts $x_t$ to machines $i = 1, .., M$
    **for** $i = 1$ **to** $M$ **in parallel do**
        Sample $z_{t,i}$ from local dataset $\mathcal{D}_i$
        Compute $v_{t,i} = \nabla f_i(x_t, z_{t,i})$
        Sample $l \sim p^l$
        Compress $g_{t,i}^l = C^l(v_{t,i}), g_{t,i}^{l-1} = C^{l-1}(v_{t,i})$
        Construct $\tilde{g}_{t,i} = g_{t,i}^0 + \frac{1}{p^l}(g_{t,i}^l - g_{t,i}^{l-1})$
        Send $\tilde{g}_{t,i}$ to server
    **end for**
    Server aggregates: $\tilde{g}_t = \frac{1}{M}\sum_{i=1}^M \tilde{g}_{t,i}$
    Server updates: $x_{t+1} = x_t - \eta\tilde{g}_t$
**end for**

---

Note that the optimization scheme in Alg. 2 is very similar to that of Alg. 1 (regular data-parallel SGD). That is thanks to the unbiasedness of the MLMC estimates (Lemma 3.2).

In the next subsections, we analyze our algorithm and derive the optimal level probabilities that minimize the MLMC estimate variance for popular baseline compressors, and for special cases of gradient distributions that arise in deep learning models.

### 3.1. MLMC-Compression Using Bit-Wise Compressors

A popular compression method used in distributed learning is bit-wise compression, especially *fixed-point* and *floating-point* compression (Seide et al., 2014; Dryden et al., 2016; Chmiel et al., 2021). We now discuss the *fixed-point*-based MLMC compression scheme. The analysis of *floating-point* MLMC compression is similar but does not enjoy the same compression rate since the *exponent* must always be transmitted. We defer the full analysis to App. B.

Since fixed-point compressors operate in an element-wise manner, we consider some entry $e_{t,i}$ of a gradient vector $v_{t,i}$. Assuming $|e_{t,i}| \leq 1$ (note that we can divide the entries of $v_{t,i}$ by the largest entry and transmit it as well), $e_{t,i}$ can be written as a 64-bit *fixed-point* binary number, as follows:

$$e_{t,i} = (-1)^{b_0} \sum_{j=1}^{63} b_j 2^{-j}, \qquad (7)$$

where $b_j \in \{0, 1\}$ is the $j$-th bit in the binary representation. For each entry $e_{t,i}$, the multilevel fixed-point compressor $C^l$ truncates the sum to $l$ elements, with $l$ ranging between 1 and 63. The resulting distortion introduced by the compression is bounded by $2^{-l}$ for each entry.

We incorporate the fixed-point compressor into the MLMC compression scheme. Each entry of the residual $g_{t,i}^l - g_{t,i}^{l-1}$ in this case consists of two bits, one information bit and one sign bit. Therefore, the transmission cost of the MLMC gradient, $\tilde{g}_{t,i}$, is the cost of transmitting two bits for each entry in the residual vector, 64 additional bits for the maximum entry, and $\lceil \log_2(63) \rceil$ for $l$, i.e., $2d + 64 + \lceil \log_2(63) \rceil$ bits in total in each iteration for each machine. Note that when $d \gg 1$ (which is often the case in deep learning), this compression scheme transmits approximately $2d$ bits in each iteration, compared to $64d$ bits for the uncompressed vectors. This is a $\times 32$ improvement in communication costs. Furthermore, the variance-minimizing level probabilities are formalized in Lemma 3.3 (proof in App. C).

**Lemma 3.3.** *The optimal probability distribution that minimizes the variance of the fixed-point MLMC estimator is given by:*

$$p^l = \frac{2^{-l}}{1 - 2^{-63}}. \qquad (8)$$

### 3.2. MLMC-Compression Using Top-$k$

Given any vector $v \in \mathbb{R}^d$, Top-$k$ retains its largest $k$ elements (in absolute value) and zeros the rest. Namely, Top-$k$ is a biased compressor with $\alpha = k/d$, whose distortion satisfies the following bound for any vector $v \in \mathbb{R}^d$ (note that Top-$k$ is deterministic):

$$\|C(v) - v\|^2 \leq (1 - \alpha) \|v\|^2 \qquad (9)$$

Similarly to the analysis with bit-wise compressors, we wish to find the optimal probability distribution $p^l$ over the compression levels $l \in [L]$. However, note that the bound in Eq. (9) is a worst-case bound, and the equality is satisfied only when $v$ is uniform. Fortunately, in practice and especially in Deep Learning, we often encounter non-uniform gradients (Glorot & Bengio, 2010). This key observation serves as motivation for developing more adaptive methods to close this gap.

We exploit this often-overlooked property and use a tighter *adaptive* bound for each sample to further enhance our

method. For a given vector $v_{t,i} \in \mathbb{R}^d$, the distortion introduced by Top-$k$ and some compression level $l \in [L]$ can be written as follows:

$$\|C^l(v_{t,i}) - v_{t,i}\|^2 = (1 - \alpha_{t,i}^l) \|v_{t,i}\|^2 \qquad (10)$$

where $0 < \alpha_{t,i}^l \leq 1$ is chosen appropriately such that the equality is satisfied. Eq. (10) describes the tightest possible bound (an equality) on the distortion introduced by the compressor, and this bound is different (adaptive) for different vectors $v_{t,i}$.

Additionally, note that when using Top-$k$ with our method in Alg. 2, the residual $g_{t,i}^l - g_{t,i}^{l-1}$ consists only of one term that corresponds to the $l$'th largest element (in absolute value) of the uncompressed stochastic gradient $v_{t,i}$. Thus, the communication cost in this case will be the cost of transmitting one entry. Similarly, for $s$-Top-$k$, $g_{t,i}^l - g_{t,i}^{l-1}$ consists of the $l$'th largest segment (in norm) of $v_{t,i}$ (containing $s$ entries, at most), thus the communication cost will be that of transmitting $s$ numbers.

Given this insight, in Lemma 3.4, we exploit this adaptive bound and use it to derive an adaptive probability distribution over the compression levels that minimizes the variance of the MLMC gradient in each iteration.

**Lemma 3.4.** *Given any multilevel compressor $C^l$, the optimal probability distribution that minimizes the variance of MLMC estimator in iteration $t \in [T]$ and for machine $i \in [M]$ is given by:*

$$p_{t,i}^l = \frac{\Delta_{t,i}^l}{\sum_{l'=1}^{L} \Delta_{t,i}^{l'}} \qquad (11)$$

where $\Delta_{t,i}^l = \|g_{t,i}^l - g_{t,i}^{l-1}\|$, and note that for a multilevel compressor based on $s$-Top-$k$, $p_{t,i}^l$ in Lemma 3.4 further reduces to $p_{t,i}^l = \frac{\sqrt{\alpha_{t,i}^l - \alpha_{t,i}^{l-1}}}{\sum_{l'=1}^{L} \sqrt{\alpha_{t,i}^{l'} - \alpha_{t,i}^{l'-1}}}$ (proof in App. D).

We incorporate this adaptive probability distribution over the compression levels with our MLMC compression method into a new adaptive optimization scheme formalized in Alg. 3. This optimization scheme is similar to that of Alg. 2, although here, the level probability distribution is chosen in an *adaptive* manner for each sample in each step (see Lemma 3.4). Since the MLMC gradients are unbiased estimates of the true gradients, namely $\mathbb{E}[\tilde{g}_{t,i}|x_t] = \nabla f_i(x_t), \forall t \in [T], \forall i \in [M]$, only the variance term in Theorem 2.3 will be affected.

Interestingly, our method recovers importance sampling (IS) techniques (e.g., (Beznosikov et al., 2020)) in certain cases. For example, in the case of Top-$k$, our method is equivalent to sampling and communicating the $l$-th entry of $v_{t,i}$ (scaled by $1/p_{t,i}^l$) with probability $p_{t,i}^l$. However, our

---

**Algorithm 3** Adaptive MLMC-Compressed Parallel SGD

---

**Input:** initialization $x_1$, step-size $\eta$, multilevel compressors $\{C^l\}_{l=1}^L$

**for** $t = 1$ **to** $T$ **do**

    The server broadcasts $x_t$ to machines $i = 1, .., M$

    **for** $i = 1$ **to** $M$ **in parallel do**

        Sample $z_{t,i}$ from local dataset $\mathcal{D}_i$

        Compute $v_{t,i} = \nabla f_i(x_t, z_{t,i})$

        Compute $p_{t,i}^l = \frac{\Delta_{t,i}^l}{\sum_{l'=1}^L \Delta_{t,i}^{l'}}$ for $l \in [L]$

        Sample $l \sim p_{t,i}^l$

        Compress $g_{t,i}^l = C^l(v_{t,i})$, $g_{t,i}^{l-1} = C^{l-1}(v_{t,i})$

        Construct $\tilde{g}_{t,i} = g_{t,i}^0 + \frac{1}{p^l}(g_{t,i}^l - g_{t,i}^{l-1})$

        Send $\tilde{g}_{t,i}$ to server

    **end for**

    Server aggregates: $\tilde{g}_t = \frac{1}{M}\sum_{i=1}^M \tilde{g}_{t,i}$

    Server updates: $x_{t+1} = x_t - \eta\tilde{g}_t$

**end for**

---

method *strictly generalizes* IS techniques, as it is compatible with complex structured compressors that do not admit such a coordinate-wise decomposition, and where IS is not naturally defined. This equivalence seems to arise only for sparsification-based methods such as Top-$k$. More involved compression methods exist for which there are no IS-like interpretations, e.g., structured quantization-based methods such as Round-to-Nearest (RTN) (Gupta et al., 2023) and ECUQ (Dorfman et al., 2023).

RTN-based methods, for example, quantize each element of a given vector $v$ by rounding it to the nearest level on a fixed grid. The spacing of this grid is controlled by a quantization step-size $\delta^l$. Namely, the RTN-compression (of level-$l$) of $v$ is given by $C_{RTN}^l(v) = \delta^l \cdot \text{clip}(\text{round}(v/\delta^l), -c, c)$, where $\delta^l = \frac{2c}{2^l - 1}$ and "round" rounds each element to its nearest integer. Here, $l \in \mathbb{N}$ corresponds to the compression level. No IS interpretation exists in this case since the difference $g_{t,i}^l - g_{t,i}^{l-1}$ does not necessarily reduce to a simple structure that can facilitate IS.

Moreover, IS requires a specific, nontrivial construction which differs for each compression method, whereas our MLMC compression functions as a plug-and-play framework that works for any series of compressors satisfying Definition 3.1, without requiring any additional tuning or specific construction.

### 3.3. Special Case Analysis

Our adaptive MLMC compression scheme seems especially attractive in scenarios in which the entries of the gradients are far from uniform. Interestingly, in many cases when training deep learning models, the gradients appear to have special structures that we can exploit (Micikevicius et al.,

2018). Specifically, (Glorot & Bengio, 2010; Shi et al., 2019) show that gradients in neural networks during training often have Gaussian-like distributions. We demonstrate the adaptability of our method in this case and show that it indeed exploits this special structure for more efficient training. For ease of analysis, let us consider a more relaxed case in which the entries of the gradients decay exponentially in absolute value (note that $ae^{-x^2} \le be^{-x}, \forall a, x \in \mathbb{R}$, for an appropriate choice of $b$, and therefore this is the more general case). We formalize this in Assumption 3.5.

**Assumption 3.5.** For any $t \in [T]$ and any $i \in [M]$, the sorted entries of the gradient $v_{t,i}$ satisfy, for $r_{t,i} > 0$:

$$|v_{t,i}(j)| = |v_{t,i}(0)|e^{-\frac{r_{t,i}}{2}j}$$

Note that this assumption implies that most of the energy of $v_{t,i}$ is concentrated in $\approx 1/r_{t,i}$ entries. This observation gives rise to two regimes depending on the relative values of $1/r_{t,i}$ and the length of the vector $d$: (1) $d$ is very small compared to $1/r_{t,i}$, which implies slow decay and the tail is not negligible. If decay is very slow, i.e., the entries are approximately uniform, our method, Rand-$k$, and Top-$k$ perform similarly; and (2) $d$ is very large compared to $1/r_{t,i}$, which implies that a the tail of the gradient vector is negligible. Here, we expect our method to have a significant benefit over other unbiased estimators (e.g., Rand-$k$). This is the more interesting case and we formalize it in Lemma 3.6 (Please refer to App. E for the full proof).

**Lemma 3.6.** *Under Assumption 3.5 for sufficiently large* $r \cdot d$, *Alg. 3 with the $s$-Top-$k$ compressor, and the optimal probabilities in Lemma 3.4, guarantees* $\mathcal{O}\left(\frac{1}{r_{t,i}s}\right)$ *variance of the MLMC estimator.*

In contrast, the variance of the compressed gradients when using Rand-$k$ with $k = s$ is $\mathcal{O}\left(\frac{d}{s}\right)$ (Condat et al., 2022). Thus, when $1/r_{t,i} < d$, our MLMC compressor enjoys smaller variance.

## 4. Convergence and Parallelization

We proposed a novel method that bridges the strengths of biased and unbiased methods by leveraging MLMC techniques to generate unbiased estimates of biased-compressed gradients. Our method statistically retains the more important parts of the gradients (similar to biased compression methods) while still enjoying good parallelization guarantees (like unbiased methods).

Note that since our MLMC gradient estimates are unbiased, a similar error bound to Eq. (2) holds, with an additional term that stems from compression. For simplicity, we focus on the homogeneous data setting. We formalize this in the following Theorem (we defer the proof to App. F.1).

**Theorem 4.1.** *Under Assumptions 2.1-2.2, Alg. 2 and Alg. 3 guarantee the following error bounds in the homogeneous convex and nonconvex cases, respectively:*

$$\mathbb{E}[f(\bar{x}_T) - f(x^*)] \in \mathcal{O}\left(\frac{D^2 L}{T} + \frac{\hat{\omega}^2 D^2 L}{MT} + \frac{(\hat{\omega}+1)\sigma D}{\sqrt{MT}}\right)$$

$$\frac{1}{T}\sum_{t=1}^{T}\mathbb{E}\|\nabla f(x_t)\|^2 \in \mathcal{O}\left(\frac{\Delta_1 L}{T} + \frac{\hat{\omega}^2 \Delta_1 L}{MT} + \frac{(\hat{\omega}+1)\sigma\sqrt{L}}{\sqrt{MT}}\right)$$

where $\hat{\omega}$ is the compression coefficient of our MLMC estimator (see Eq. (3)). Exact calculations of $\hat{\omega}$ for various compressors are available in App. B, D, E. Note that the middle term is asymptotically negligible compared to the right term, and thus these error bounds are asymptotically identical to those of Parallel-SGD (Theorem 2.3; Eq. (2)), with a slightly increased variance due to compression.

In contrast, the error bound for biased methods, e.g., EF21-SGDM (Corollary 3 in (Fatkhullin et al., 2023)), which is the current state of the art, is given by (nonconvex case):

$$\frac{1}{T}\sum_{t=1}^{T}\mathbb{E}\|\nabla f(x_t)\|^2 \in \mathcal{O}\left(\frac{\Delta_1 L}{\alpha T} + \frac{\Delta_1 L \sigma^{1/2}}{\alpha^{1/2}T^{3/4}} + \frac{\Delta_1 L \sigma}{\sqrt{MT}}\right)$$

Thus, our method allows parallelization over $M = \mathcal{O}(T)$, or equivalently $M = \mathcal{O}(\sqrt{N})$, machines without a degradation of performance (where $N$ is the size of the dataset), while EF21-SGDM allows $M = \mathcal{O}(\sqrt{T})$, or equivalently $\mathcal{O}(N^{1/3})$. Moreover, our method complements methods like EF21-SGDM and others (which may be beneficial when $M$ is small), in the regime of *massive* parallelization, i.e., when $M$ is very large. We defer the analysis to App. F.3.

Our method works in the heterogeneous data setting as well, although a $\mathcal{O}\left(\frac{\hat{\omega}\xi}{\sqrt{MT}}\right)$ term is added to the error bounds (in the convex and nonconvex cases), where $\xi \geq 0$ quantifies the heterogeneity $\|\nabla f_i(x) - \nabla f(x)\|^2 \leq \xi^2, \forall x \in \mathbb{R}^d$. Please refer to App. F.4 for the full analysis. Moreover, since our MLMC compression method produces unbiased gradient estimates, it can be seamlessly incorporated into more sophisticated optimization templates such as MARINA (Gorbunov et al., 2021) or DASHA (Tyurin & Richtárik, 2023), which would fully mitigate the heterogeneity term.

## 5. Experiments

We present several deep learning experiments involving finetuning BERT (Devlin et al., 2018) on GLUE SST-2 (Wang et al., 2018) and CIFAR-10 (Krizhevsky, 2009) image classification using ResNet18 (He et al., 2016). We evaluated the performance of our MLMC-based compressors in comparison to biased and unbiased compressors. Our experiments were implemented using PyTorch and executed on NVIDIA GeForce RTX 4090 GPUs.

### 5.1. Experiments with Sparsification Compressors

In the first set of experiments, we tested our MLMC-compression technique, with Top-$k$ as a baseline compressor, and optimized the learning rate for each one individually. We compared the performance of our **Adaptive MLMC-Top-$k$** compressor (Alg. 3), the biased compressors **Top-$k$** and **EF21-SGDM** (Fatkhullin et al., 2023), and the unbiased compressor **Rand-$k$**, and **Uncompressed SGD** as a baseline. We evaluated two criteria: *communication efficiency* and *iteration efficiency*, which compare the test accuracy of the algorithms as a function of the communication complexity (the number of communicated bits) and as a function of the epochs (the number of iterations).

We present the *communication efficiency* experimental results in Figure 1, for $M = 4$ machines (top quartet) and $M = 32$ machines (bottom quartet). Each subplot displays the test accuracy of the compared algorithms, against the number of communicated bits, for various sparsification levels, specifically for $k \in \{0.01n, 0.05n, 0.1n, 0.5n\}$, where $n \approx 1.1 \times 10^8$ is the number of model parameters. We used a batch size of 16 in all experiments and averaged over 5 seeds. Moreover, we display these results against the number of epochs (iterations) in Figure 2.

Figures 1-2 show that our MLMC-compression method outperforms the other methods, both in terms of communication and iteration efficiency. Notably, our method achieves a higher test accuracy for the same number of transmitted bits and enjoys a faster convergence rate compared to other methods across different sparsification levels. Also, our method converges faster for $M = 32$ compared to $M = 4$, which is consistent with our bounds in Theorem 4.1. Moreover, Figure 2 shows that our method outperforms other compression methods in iteration efficiency, in terms of convergence rate and accuracy, and enjoys the same performance as *uncompressed* SGD, despite using significantly less information. Additional experiments on CIFAR-10 image classification using ResNet18 are available in App. G.1.

### 5.2. Experiments with Bit-Wise Quantization

We evaluated our nonadaptive MLMC-compression method (Alg. 2) with bit-wise quantization compressors on image classification tasks using the ResNet18 architecture and the CIFAR-10 dataset. We compare the communication efficiency of our method to biased **2-bit quantization**, unbiased **2-bit QSGD** (Alistarh et al., 2017), for the same compression level, and **uncompressed SGD** as a baseline. We present the results in Figure 3. These results show that also in the case of bit-wise compressors, our method enjoys a significant advantage over the others in terms of communication efficiency, convergence rate, and final test accuracy. Additional experiments evaluating RTN compressors on BERT GLUE SST2 finetuning are available in App. G.2.

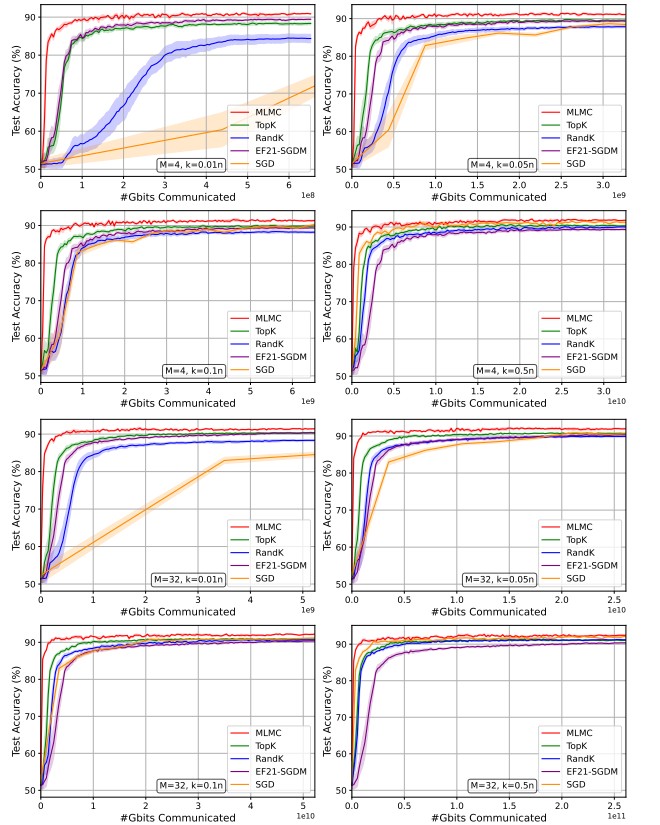

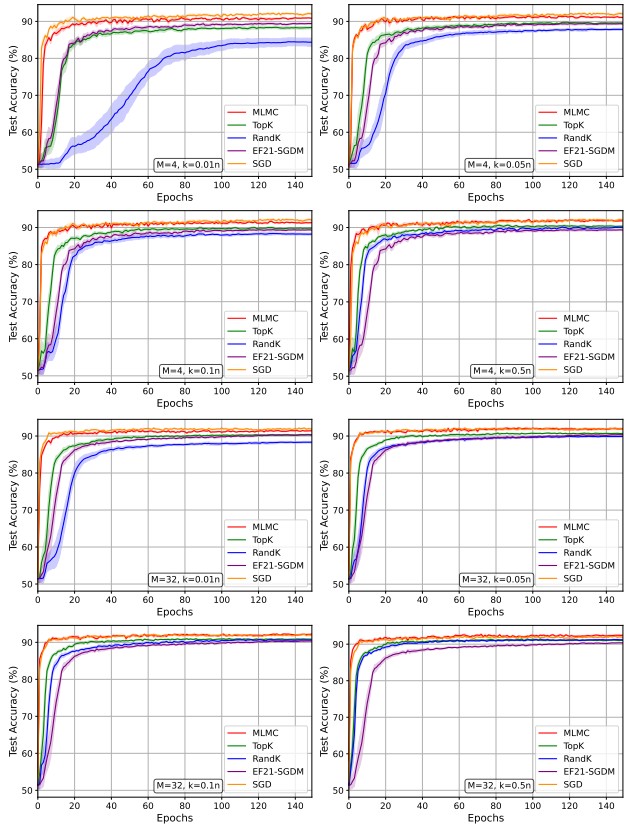

*Figure 1.* Finetuning BERT on GLUE SST2 **communication efficiency** comparison of the Adaptive MLMC-Top-$k$ (Alg. 3), Top-$k$, EF21-SGDM, and Rand-$k$, and uncompressed SGD for sparsification levels $k \in \{0.01n, 0.05n, 0.1n, 0.5n\}$, $M = 4, 32$ machines, and a batch size of 16 samples, averaged over 5 different seeds.

*Figure 2.* Finetuning BERT on GLUE SST2 **iteration efficiency** comparison of the Adaptive MLMC-Top-$k$ (Alg. 3), Top-$k$, EF21-SGDM, Rand-$k$, and uncompressed SGD for sparsification levels $k \in \{0.01n, 0.05n, 0.1n, 0.5n\}$, $M = 4, 32$ machines, and a batch size of 16 samples, averaged over 5 different seeds.

## 6. Conclusions

We presented a novel method that bridges the gap between unbiased and biased compression approaches typically used to overcome communication overhead in distributed learning settings. MLMC serves at the heart of our method and facilitates the transduction of bias into variance, combining the strengths of both worlds: the superior empirical performance of biased methods and the strong theoretical guarantees of unbiased techniques. We validated our algorithms on deep learning tasks showcasing their empirical efficiency compared to existing methods.

## Acknowledgments

This research was partially supported by Israel PBC-VATAT, by the Technion Artificial Intelligent Hub (Tech.AI), and by the Israel Science Foundation (grant No. 3109/24). The second author would like to thank VATAT (through the Israel Council for Higher Education) for supporting this research.

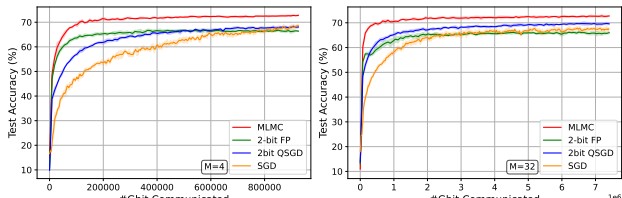

*Figure 3.* CIFAR-10 image classification using ResNet18, communication efficiency comparison of our Fixed-Point-based MLMC compression method (Alg. 2), 2-bit Fixed-Point quantization, 2-bit QSGD, and uncompressed SGD, for $M = 4$ machines and a batch size of 128 and for $M = 32$ machines and a batch size of 64, averaged over 5 different seeds.

## Impact Statement

This paper presents work whose goal is to advance the field of Machine Learning. There are many potential societal consequences of our work, none which we feel must be specifically highlighted here.

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

## A. Proof of Lemma 3.2

**Lemma 3.2** *For any multilevel compressor $C^l, l \in [L]$, any non-zero level probabilities $\{p^l\}_{l=1}^{L}$, the MLMC gradient estimator $\tilde{g}_{t,i} \triangleq g_{t,i}^0 + \frac{1}{p^l}(g_{t,i}^l - g_{t,i}^{l-1})$ is a conditionally unbiased estimate of the true gradient at step $t$, $\nabla f_i(x_t), \forall t \in [T], \forall i \in [M]$. Namely: $\mathbb{E}[\tilde{g}_{t,i}|x_t] = \nabla f_i(x_t)$.*

*Proof.*

$$\mathbb{E}[\tilde{g}_{t,i}|x_t] = \mathbb{E}_{l \sim p^l, \, z_{t,i} \sim \mathcal{D}_i}[g_{t,i}^0 + \frac{1}{p^l}(g_{t,i}^l - g_{t,i}^{l-1})|x_t] \tag{12}$$

$$= \mathbb{E}_{z_{t,i} \sim \mathcal{D}_i}[\mathbb{E}_{l \sim p^l}[g_{t,i}^0 + \frac{1}{p^l}(g_{t,i}^l - g_{t,i}^{l-1})|x_t, z_{t,i}]] \tag{13}$$

$$= \mathbb{E}_{z_{t,i} \sim \mathcal{D}_i}[\sum_{l=1}^{L} p^l(g_{t,i}^0 + \frac{1}{p^l}(g_{t,i}^l - g_{t,i}^{l-1}))|x_t] \tag{14}$$

$$\overset{(1)}{=} \mathbb{E}_{z_{t,i} \sim \mathcal{D}_i}[g_{t,i}^0 + \sum_{l=1}^{L}(g_{t,i}^l - g_{t,i}^{l-1})|x_t] \tag{15}$$

$$= \mathbb{E}_{z_{t,i} \sim \mathcal{D}_i}[g_{t,i}^L|x_t] \tag{16}$$

$$\overset{(2)}{=} \mathbb{E}_{z_{t,i} \sim \mathcal{D}_i}[\nabla f_i(x_t, z_{t,i})|x_t] \tag{17}$$

$$= \nabla f_i(x_t) \tag{18}$$

where transition (1) since $\{p^l\}_{l=1}^{L}$ is a probability distribution, i.e., $\sum_{l=1}^{L} p^l = 1$. (2) follows since by the definition of multilevel compressors in 3.1 where the highest level $L$ corresponds to no compression (e.g., top-$k$ with $k = d$).

## B. Analysis of the Floating-Point based MLMC compressor

Given an element of the gradient $v$ denoted by $e$, it can be represented as a 64-bit *floating-point* binary number. The floating-point number consists of three parts - the *Sign* denoted by $S$, the *Exponent* denoted by $E$ and the *Mantissa* which is a binary number with digits $\{m_i\}_{i=1}^{52}$. The entry $e$ can be written as follows:

$$e = (-1)^S 2^{E-1023}\left(1 + \sum_{j=1}^{52} m_j 2^{-j}\right) \tag{19}$$

The Floating-Point Compressor $C^l(e)$ truncates the sum to $l$ elements, which implies that the resolution will be up to $2^{E-1023}2^{-l}$, and the Compressor's parameter $l$ (which determines the extent of compression) ranges between 1 to 52. Since the Exponent is $E = \lfloor \log_2(e) \rfloor + 1023$, the Floating-Point biased compressor satisfies Eq. (4) with $\alpha = 1 - 2^{-l}$.

We apply the MLMC scheme with the Floating-Point Compressor and thus only need to transmit the residual $g_{t,i}^l - g_{t,i}^{l-1}$. The residual has the same Exponent and Sign bits as the original entry, but contains only *one* information bit of the mantissa for every element in the vector. This means that the Floating-Point MLMC compressor needs to only transmit $13d + \log_2(52)$ bits instead of $64d$ bits, where the extra $\log_2(52)$ bits are needed to transmit the sampled $l$. Furthermore, note that for $d \gg 1$ those additional bits are negligible, implying $\times \frac{64}{13} \approx \times 4.9$ improvement in Communication cost.

Similarly to the Fixed-Point case, the MLMC technique requires a probability distribution to sample the $l$-compression parameter with the compressor. We would like to use the ideal distribution to minimize the variance introduced by the compression process. We formalize this in Lemma B.1.

**Lemma B.1.** *The optimal probability distribution that minimizes the variance of the Floating-Point MLMC estimator is given by:*

$$p^l = \frac{2^{-l}}{1 - 2^{-52}} \tag{20}$$

*Proof.* The second moment of the Floating-Point MLMC compressor is given by:

$$\mathbb{E}[\|\tilde{g}_{t,i}\|^2] = \mathbb{E}\left[\left\|g_{t,i}^0 + \frac{1}{p^l}(g_{t,i}^l - g_{t,i}^{l-1})\right\|^2\right] \tag{21}$$

where $\|\cdot\|$ is the $l_2$-norm. Since this compressor operates in an element-wise manner, we consider the $r$-th element in $\tilde{g}_{t,i}$, which we denote by $\tilde{e}_{t,i}^2(r)$. Since $e_{t,i}^0(r) = 0$ for every $r$ we obtain:

$$\mathbb{E}\left[\left\|\frac{1}{p^l}(e_{t,i}^l(r) - e_{t,i}^{l-1}(r))\right\|^2\right] \overset{(1)}{=} \sum_{l=1}^{52}\left[p_l\left|\frac{1}{p^l}\left(2^{E(r)-1023}\left(1 + \sum_{j=1}^{l}m_j(r)2^{-j}\right) - 2^{E(r)-1023}\left(1 + \sum_{i=1}^{l-1}m_i(r)2^{-i}\right)\right)\right|^2\right] \tag{22}$$

$$= \sum_{l=1}^{52}\left[\frac{2^{2E(r)-2026}}{p^l}\left(m_l(r)2^{-l}\right)^2\right] \tag{23}$$

$$\overset{(2)}{=} \sum_{l=1}^{52}\left[\frac{2^{2E(r)-2026}}{p^l}m_l(r)2^{-2l}\right] \tag{24}$$

where (1) follows by the floating-point representation of the $e_{t,i}^l(r)$, and (2) follows since every binary digit $m_l$ satisfies $m_l^2 = m_l$ (since $m^l$ can either be 0 or 1). Now, similarly to the proof in appendix C, we wish to find the optimal probability distribution that minimizes the variance (note that the probability distribution should sum to 1). There are no additional assumptions on the binary number we wish to compress. Namely, for any $l$, $m_l$ can be 1 or 0, and we would like to minimize the objective regardless of the values of $m^l$. We formalize this using the following optimization problem:

$$\hat{p}^l = \underset{\{p^l\}_{l=1}^{52}}{\arg\min}\max_{\lambda \geq 0}\sum_{l=1}^{52}\left[\frac{2^{2E(r)-2026}}{p^l}2^{-2l}\right] + \lambda\left(\sum_{l=1}^{52}p^l - 1\right) \tag{25}$$

By setting the gradients with respect to $p^l$ and $\lambda$ to zero, we obtain the following:

$$\sum_{l=1}^{52}\hat{p}^l = 1 \tag{26}$$

$$\hat{p}^l = \frac{2^{E(r)-1023}}{\sqrt{\lambda}}2^{-l} \tag{27}$$

These equations show that $\hat{p}^l$ is proportional to $2^{-l}$. Thus, with proper normalization, by solving for $\lambda$ and extracting $\hat{p}^l$, we have:

$$\hat{p}^l = \frac{2^{-l}}{1 - 2^{-52}} \tag{28}$$

which concludes our proof. $\qquad\square$

Note that we can calculate the optimal variance of the MLMC estimator using the optimal probabilities we obtained above. We first calculate the second moment of some element in the MLMC gradient estimate:

$$\mathbb{E}[\|\tilde{e}_{t,i}(r)\|^2] = \mathbb{E}\left[\left\|\frac{1}{p^l}(e_{t,i}^l(r) - e_{t,i}^{l-1}(r))\right\|^2\right] = \sum_{l=1}^{52}\left[\frac{2^{2E(r)-2026}}{p^l}m_l(r)2^{-2l}\right] = \sum_{l=1}^{52}\left[2^{2E(r)-2026}(1 - 2^{-52})m_l(r)2^{-l}\right] \tag{29}$$

$$= 2^{E(r)-1023}(1 - 2^{-52})\left((2^{E(r)-1023})\left(1 + \sum_{l=1}^{52}\left[m_l(r)2^{-l}\right]\right) - (2^{E(r)-1023})\right) \tag{30}$$

$$= 2^{E(r)-1023}(1 - 2^{-52})(e(r) - (2^{E(r)-1023})) \tag{31}$$

Now, by summing the second moments of all the elements and using the unbiasedness of the MLMC estimator (Lemma 3.2), we obtain the compression variance component of the MLMC estimator's variance as follows (Note that the total variance is given by $\sigma_{comp}^2 + \sigma^2$):

$$\sigma_{comp}^2 = \mathbb{E}[\|\tilde{g}_{t,i}\|^2] - (\mathbb{E}[\|\tilde{g}_{t,i}\|])^2 = \sum_{r=1}^{d} \mathbb{E}[\|\tilde{e}_{t,i}(r)\|^2] - v_{t,i}^2 \tag{32}$$

$$= \sum_{r=1}^{d} \left[ 2^{E(r)-1023}(1 - 2^{-52})(e(r) - (2^{E(r)-1023})) \right] - v_{t,i}^2 \tag{33}$$

## C. Proof of Lemma 3.3

**Lemma 3.3** *The optimal probability distribution that minimizes the variance of the Fixed-Point MLMC estimator is given by:*

$$p^l = \frac{2^{-l}}{1 - 2^{-63}} \tag{34}$$

*Proof.* The second moment of the Fixed-Point MLMC compressor is given by:

$$\mathbb{E}[\|\tilde{g}_{t,i}\|^2] = \mathbb{E}\left[ \left\| g_{t,i}^0 + \frac{1}{p^l}(g_{t,i}^l - g_{t,i}^{l-1}) \right\|^2 \right] \tag{35}$$

where $\|\cdot\|$ is the $l_2$-norm. Since fixed-point compressors are element-wise, similarly to the floating-point compressor, we consider a single entry of $\tilde{g}_{t,i}$, which we denote by $\tilde{e}_{t,i}^2$. Since $e_{t,i}^0 = 0$, we have:

$$\mathbb{E}\left[ \left| \frac{1}{p^l}(e_{t,i}^l - e_{t,i}^{l-1}) \right|^2 \right] \overset{(1)}{=} \sum_{l=1}^{63} \left[ p_l \left| \frac{1}{p^l} \left( \sum_{j=1}^{l} b_j 2^{-j} - \sum_{i=1}^{l-1} b_i 2^{-i} \right) \right|^2 \right] \tag{36}$$

$$= \sum_{l=1}^{63} \left[ \frac{1}{p^l} \left( b_l 2^{-l} \right)^2 \right] \tag{37}$$

$$\overset{(2)}{=} \sum_{l=1}^{63} \left[ \frac{1}{p^l} b_l 2^{-2l} \right] \tag{38}$$

where (1) follows using the binary representation of the normalized element. and (2) follows since every binary $b_l^2 = b_l$ (note that $b^l$ can only be 0 or 1). We wish to find the optimal probability distribution that minimizes the variance (note that the probability distribution should sum to 1). There are no additional assumptions on the binary number we wish to compress. Namely, for any $l$, $b_l$ can be 1 or 0, and we would like to minimize the objective regardless of the values of $b^l$. We bound $b^l$ and formalize this in the following optimization problem:

$$\hat{p}^l = \arg\min_{\{p^l\}_{l=1}^{63}} \max_{\lambda \geq 0} \sum_{l=1}^{63} \left[ \frac{1}{p^l} 2^{-2l} \right] + \lambda \left( \sum_{l=1}^{63} p^l - 1 \right) \tag{39}$$

by setting the gradients with respect to $p^l$ and $\lambda$ to zero, we obtain the following:

$$\sum_{l=1}^{63} \hat{p}^l = 1 \tag{40}$$

$$\hat{p}^l = \frac{1}{\sqrt{\lambda}} 2^{-l} \tag{41}$$

which imply that $\hat{p}^l$ must be proportional to $2^{-l}$, and with with proper normalization (by solving for $\lambda$ and extracting $p^l$), we have:

$$\hat{p}^l = \frac{2^{-l}}{1 - 2^{-63}} \tag{42}$$

which concludes the proof. $\square$

Note that we can calculate the variance of the MLMC estimator using the optimal probabilities that we obtained. We start by calculating the second moment of some entry in the MLMC gradient estimate vector $\tilde{g}_{t,i}$.

$$\mathbb{E}[\|\tilde{e}_{t,i}\|^2] = \mathbb{E}\left[\left\|\frac{1}{p^l}(e_{t,i}^l - e_{t,i}^{l-1})\right\|^2\right] = \sum_{l=1}^{63}\left[\frac{1}{p^l}b_l 2^{-2l}\right] = (1 - 2^{-63})\sum_{l=1}^{63}\left[b_l 2^{-l}\right] = (1 - 2^{-63})|e_{t,i}| \approx |e_{t,i}| \quad (43)$$

and by the unbiasedness of the MLMC estimate (Lemma 3.2), its compression variance is given by (note that the total variance is equal to $\sigma_{comp}^2 + \sigma^2$):

$$\sigma_{comp}^2 = \mathbb{E}[\|\tilde{g}_{t,i}\|^2] - (\mathbb{E}[\|\tilde{g}_{t,i}\|])^2 = (1 - 2^{-63})\|v_{t,i}\|_1 - \|v_{t,i}\|^2 \quad (44)$$

## D. Proof of Lemma 3.4

**Lemma 3.4** *Given any multilevel compressor $C^l$, the optimal probability distribution that minimizes the variance of MLMC estimator in iteration $t \in [T]$ and for machine $i \in [M]$ is given by:*

$$p_{t,i}^l = \frac{\Delta_{t,i}^l}{\sum_{l'=1}^{L}\Delta_{t,i}^{l'}}, \quad (45)$$

*where $\Delta_{t,i}^l$ is the $\ell_2$ norm of the residual vector at step t, i.e., $\Delta_{t,i}^l = \left\|g_{t,i}^l - g_{t,i}^{l-1}\right\|$.*

*Proof.* The second moment of the MLMC-based compressor is given by:

$$\mathbb{E}[\|\tilde{g}_{t,i}\|^2] = \mathbb{E}\left[\left\|g_{t,i}^0 + \frac{1}{p_{t,i}^l}(g_{t,i}^l - g_{t,i}^{l-1})\right\|^2\right] \quad (46)$$

Using our definition that $g_{t,i}^0 = 0$ and the definition of $\Delta_{t,i}^l$, we obtain:

$$\mathbb{E}[\|\tilde{g}_{t,i}\|^2] = \mathbb{E}\left[\frac{1}{(p_{t,i}^l)^2}(\Delta_{t,i}^l)^2\right] \quad (47)$$

amd by writing the expectation w.r.t $p_l$ explicitly, we have:

$$\mathbb{E}[\|\tilde{g}_{t,i}\|^2] = \sum_{l=1}^{L}\left[\frac{1}{p_{t,i}^l}(\Delta_{t,i}^l)^2\right] \quad (48)$$

We wish to find the optimal probability distribution that minimizes the variance (note that the probability distribution should sum to 1). We formalize this into the following optimization problem:

$$\hat{p}_{t,i}^l = \arg\min_{\{p_{t,i}^l\}_{l=1}^{L}} \max_{\lambda \geq 0} \sum_{l=1}^{L}\left[\frac{1}{p_{t,i}^l}(\Delta_{t,i}^l)^2\right] + \lambda\left(\sum_{l=1}^{L}p_{t,i}^l - 1\right) \quad (49)$$

By setting the gradients with respect to $p_{t,i}^l$ and $\lambda$ to zero, we obtain the following:

$$\sum_{l=1}^{L}\hat{p}_{t,i}^l = 1 \quad (50)$$

$$\hat{p}_{t,i}^l = \frac{1}{\sqrt{\lambda}}\Delta_{t,i}^l \quad (51)$$

where $\frac{1}{\sqrt{\lambda}}$ is the normalization factor of the probability distribution. With proper normalization (by solving for $\lambda$ and extracting $\hat{p}_{t,i}^l$) the optimal probability distribution is given by:

$$\hat{p}_{t,i}^l = \frac{\Delta_{t,i}^l}{\sum_{l'=1}^{L}\Delta_{t,i}^{l'}}, \quad (52)$$

which concludes the proof. □

We calculate the variance of the MLMC estimate using the optimal probabilities we obtained. We start by writing the second moment of the MLMC gradient estimate $\tilde{g}_{t,i}$:

$$\mathbb{E}[\|\tilde{g}_{t,i}\|^2] = \mathbb{E}\left[\left\|g_{t,i}^0 + \frac{1}{p_{t,i}^l}(g_{t,i}^l - g_{t,i}^{l-1})\right\|^2\right] \tag{53}$$

$$= \sum_{l=1}^{L}\left[\frac{1}{p_{t,i}^l}(\Delta_{t,i}^l)^2\right] = \left[\sum_{l=1}^{L}\Delta_{t,i}^l\right]\cdot\left[\sum_{l'=1}^{L}\Delta_{t,i}^{l'}\right] = \left[\sum_{l=1}^{L}\Delta_{t,i}^l\right]^2 \tag{54}$$

Thus, since the MLMC estimator is unbiased (Lemma 3.2), the compression variance of the MLMC compressor is given by (note that the total variance is equal to $\sigma_{t,comp}^2 + \sigma^2$):

$$\sigma_{t,comp}^2 = \mathbb{E}[\|\tilde{g}_{t,i}\|^2] - (\mathbb{E}[\|\tilde{g}_{t,i}\|])^2 = \left[\sum_{l=1}^{L}\Delta_{t,i}^l\right]^2 - \|v_{t,i}\|^2 \tag{55}$$

This result is general and does not assume a specific multilevel compressor of the method.

Now, we apply those results to the case of $s$-Top-$k$-based MLMC compressor and derive the second moment of the MLMC estimate. Here, note that we use the adaptive distortion bound in Eq. (10) to write $\Delta_{t,i}^l$ in terms of $\alpha_{t,i}^l$. Recall that $\Delta_{t,i}^l$ is given by:

$$\Delta_{t,i}^l = \left\|g_{t,i}^l - g_{t,i}^{l-1}\right\|, \tag{56}$$

and since $g_{t,i}^l$ contains only a subset of the elements of the original uncompressed stochastic gradient $v_{t,i}$ (recall that $s$-top-$k$ retains the $k$ non-overlapping segments of length $s$ with the largest norms of the sorted stochastic gradient vector):

$$\left\|g_{t,i}^l\right\|^2 = \alpha_{t,i}^l\|v_{t,i}\|^2 \tag{57}$$

Similarly, we have:

$$(\Delta_{t,i}^l)^2 = \left\|g_{t,i}^l - g_{t,i}^{l-1}\right\|^2 = \left\|g_{t,i}^l\right\|^2 - \left\|g_{t,i}^{l-1}\right\|^2 \tag{58}$$

This is because the norm of the difference is equivalent toy the norm of the $l$-th segemnt of length $s$ in the sorted stochastic gradient. Therefore, $(\Delta_{t,i}^l)^2$ can be written as follows:

$$(\Delta_{t,i}^l)^2 = \left\|g_{t,i}^l - g_{t,i}^{l-1}\right\|^2 = \left\|g_{t,i}^l\right\|^2 - \left\|g_{t,i}^{l-1}\right\|^2 = (\alpha_{t,i}^l - \alpha_{t,i}^{l-1})\|v_{t,i}\|^2 \tag{59}$$

Plugging into the optimal probabilities and the corresponding compression variance, we have:

$$\hat{p}_{t,i}^l = \frac{\sqrt{\alpha_{t,i}^l - \alpha_{t,i}^{l-1}}}{\sum_{l'=1}^{L}\sqrt{\alpha_{t,i}^{l'} - \alpha_{t,i}^{l'-1}}} \quad ; \quad \sigma_{t,comp}^2 = \left[\left(\sum_{l=1}^{L}\sqrt{\alpha_{t,i}^l - \alpha_{t,i}^{l-1}}\right)^2 - 1\right]\|v_{t,i}\|^2 \tag{60}$$

## E. Proof of Lemma 3.6

**Lemma 3.6** *Under Assumption 3.5 for sufficiently large $r \cdot d$, Alg. 3 with the $s$-top-$k$ compressor, and the optimal probabilities in Lemma 3.4, guarantees $\mathcal{O}\left(\frac{1}{r_{t,i}s}\right)$ variance of the MLMC estimator.*

*Proof.* The compression variance of the MLMC estimator in the case of $s$-top-$k$, $\sigma_{t,comp}^2$, is derived in Appendix D and is given by (see Eq. (55)):

$$\sigma_{t,comp}^2 = \left(\sum_{l=1}^{L}\Delta_{t,i}^l\right)^2 - \|v_{t,i}\|^2 \tag{61}$$

Under Assumption 3.5, the absolute value of the $j$-th element of the uncompressed stochastic gradient $v_{t,i}$ is given by:

$$|v_{t,i}(j)| = |v_{t,i}(0)|e^{-\frac{r_{t,i}}{2}\cdot j} \tag{62}$$

Thus, the norm of the vector can be written as:

$$\|v_{t,i}\|^2 = \sum_{j=0}^{d-1} |v_{t,i}(0)|^2 e^{-r_{t,i} \cdot j} = |v_{t,i}(0)|^2 \frac{1 - e^{-r_{t,i} \cdot d}}{1 - e^{-r_{t,i}}} \tag{63}$$

where the second equality follows by the sum of a geometric series. Similarly, $(\Delta_{t,i}^l)^2$ is given by:

$$(\Delta_{t,i}^l)^2 = |v_{t,i}(0)|^2 \sum_{j=s \cdot (l-1)}^{s \cdot l - 1} e^{-r_{t,i} \cdot j} = |v_{t,i}(0)|^2 \frac{e^{-r_{t,i} \cdot s(l-1)}(1 - e^{-r_{t,i} \cdot s})}{1 - e^{-r_{t,i}}} \tag{64}$$

these results give rise to two regimes depending on the value of $r_{t,i}$ compared to $d$:

(1) $r \cdot d < 1$: In this case, the exponential decay is slow, and the "tail" of the sorted vector entries is not negligible. Namely, if decay is very slow, the gradient vector entries would be nearly uniform. This is the worst-case scenario in which our MLMC compressor, rank-$k$, and top-$k$ all have similar performance since: $\Delta_{t,i}^1 \approx \Delta_{t,i}^2 \approx ... \approx \Delta_{t,i}^L$.

(2) $r < 1$ and $r \cdot d > 1$: This is the more interesting case in which we expect our method to have an edge over the other. Accordingly, we derive an approximation for the variance under this regime. by plugging the expression of the $\Delta_{t,i}^l$ and $\|v_{t,i}\|^2$ into the expression for the variance (Eq. (55)), we have:

$$\sigma_{t,comp}^2 = |v_{t,i}(0)|^2 \left( \left( \sum_{l=1}^{L} \sqrt{\frac{e^{-r_{t,i} \cdot s(l-1)}(1 - e^{-r_{t,i} \cdot s})}{1 - e^{-r_{t,i}}}} \right)^2 - \frac{1 - e^{-r_{t,i} \cdot d}}{1 - e^{-r_{t,i}}} \right) \tag{65}$$

$$= |v_{t,i}(0)|^2 \left( \frac{1 - e^{-r_{t,i} \cdot s}}{1 - e^{-r_{t,i}}} \left( \sum_{l=1}^{L} \sqrt{e^{-r_{t,i} \cdot s(l-1)}} \right)^2 - \frac{1 - e^{-r_{t,i} \cdot d}}{1 - e^{-r_{t,i}}} \right) \tag{66}$$

$$= |v_{t,i}(0)|^2 \left( \frac{1 - e^{-r_{t,i} \cdot s}}{1 - e^{-r_{t,i}}} \left( \sum_{l=1}^{L} e^{-\frac{r_{t,i}}{2} \cdot s(l-1)} \right)^2 - \frac{1 - e^{-r_{t,i} \cdot d}}{1 - e^{-r_{t,i}}} \right) \tag{67}$$

$$= |v_{t,i}(0)|^2 \left( \frac{1 - e^{-r_{t,i} \cdot s}}{1 - e^{-r_{t,i}}} \left( \frac{1 - e^{-\frac{r_{t,i}}{2} s L}}{1 - e^{-\frac{r_{t,i}}{2} s}} \right)^2 - \frac{1 - e^{-r_{t,i} \cdot d}}{1 - e^{-r_{t,i}}} \right) \tag{68}$$

$$= |v_{t,i}(0)|^2 \left( \frac{1 - e^{-r_{t,i} \cdot s}}{1 - e^{-r_{t,i}}} \left( \frac{1 - e^{-\frac{r_{t,i}}{2} d}}{1 - e^{-\frac{r_{t,i}}{2} s}} \right)^2 - \frac{1 - e^{-r_{t,i} \cdot d}}{1 - e^{-r_{t,i}}} \right) \tag{69}$$

$$\overset{(1)}{=} \|v_{t,i}\|^2 \left( \frac{1 - e^{-r_{t,i} \cdot s}}{1 - e^{-r_{t,i} \cdot d}} \left( \frac{1 - e^{-\frac{r_{t,i}}{2} d}}{1 - e^{-\frac{r_{t,i}}{2} s}} \right)^2 - 1 \right) \tag{70}$$

where in (1) we used the expression for the norm of the gradient. To approximate the variance, we use the fact that $r \cdot d > 1$ to approximate the exponents in the expression:

$$\sigma_{t,comp}^2 = \|v_{t,i}\|^2 \left( \frac{1 - e^{-r_{t,i} \cdot s}}{1 - e^{-r_{t,i} \cdot d}} \left( \frac{1 - e^{-\frac{r_{t,i}}{2} d}}{1 - e^{-\frac{r_{t,i}}{2} s}} \right)^2 - 1 \right) \tag{71}$$

$$\approx \|v_{t,i}\|^2 \left( \frac{1 - e^{-r_{t,i} \cdot s}}{\left(1 - e^{-\frac{r_{t,i}}{2} s}\right)^2} - 1 \right) \tag{72}$$

Recall that $s$ is a hyperparameter that we can choose as we see fit, and specifically, we consider $s$ such that $s \cdot r_{t,i} \leq 1$. This implies that the number of the elements we transmit is less or equal to $\frac{1}{r_{t,i}}$. Thus, we obtain the following

approximation to the variance:

$$\sigma_{t,comp}^2 \approx \|v_{t,i}\|^2 \left( \frac{1 - e^{-r_{t,i} \cdot s}}{\left(1 - e^{-\frac{r_{t,i}}{2}s}\right)^2} - 1 \right) \tag{73}$$

$$\approx \|v_{t,i}\|^2 \left( \frac{r_{t,i} \cdot s}{\left(\frac{r_{t,i}}{2}s\right)^2} - 1 \right) \tag{74}$$

$$= \|v_{t,i}\|^2 \left( \frac{4}{r_{t,i}s} - 1 \right) \tag{75}$$

$$= \mathcal{O}\left( \frac{1}{r_{t,i}s} \right) \tag{76}$$

which concludes the proof. □

# F. Convergence and Parallelization

## F.1. Proof of Theorem 4.1

*Proof.* We assume the homogeneous data setting in which $\mathcal{D}_i \equiv \mathcal{D}$ and thus $f_i(x) = f(x), \forall i$. We analyze the convex and nonconvex cases separately.

**Homogeneous Convex case.**

Since our MLMC gradients, $\tilde{g}_{t,i}$, in Alg. 2 and Alg. 3 are unbiased estimates of the true gradients, $\nabla f(x_t)$, for all $t \in [T]$ and $i \in [M]$ (see Lemma 3.2), the following bound holds for $\eta \leq \frac{1}{2L}$ (see, e.g., Appendix A.1 in Dorfman et al. (2024)):

$$\mathbb{E}[f(\bar{x}_T) - f(x^*)] \leq \frac{1}{T}\sum_{t=1}^{T}\mathbb{E}[f(x_t) - f(x^*)] \leq \frac{D^2}{2\eta T} + \frac{\eta}{T}\sum_{t=1}^{T}\mathbb{E}V_t^2 \tag{77}$$

where $\bar{x}_T = \frac{1}{T}\sum_{t=1}^{T}$, $x^* = \arg\min_x f(x)$, $D = \|x_1 - x^*\|$, and $V_t^2 = \mathbb{E}[\|\tilde{g}_t - \nabla f(x_t)\|^2 | x_t]$. Note that in this case $V_t^2$ is the variance of the *MLMC gradients*. Let us now consider the variance term, $V_t^2$. We have:

$$V_t^2 = \mathbb{E}[\|\tilde{g}_t - \nabla f(x_t)\|^2 | x_t] \tag{78}$$

$$\overset{(1)}{=} \frac{1}{M^2}\sum_{i=1}^{M}\mathbb{E}[\|\tilde{g}_{t,i} - \nabla f(x_t)\|^2 | x_t] \tag{79}$$

$$\overset{(2)}{\leq} \frac{2}{M^2}\sum_{i=1}^{M}\left(\mathbb{E}[\|\tilde{g}_{t,i} - v_{t,i}\|^2 | x_t] + \mathbb{E}[\|v_{t,i} - \nabla f(x_t)\|^2 | x_t]\right) \tag{80}$$

$$\overset{(3)}{\leq} \frac{2}{M^2}\sum_{i=1}^{M}\left(\mathbb{E}[\|\tilde{g}_{t,i} - v_{t,i}\|^2 | x_t] + \sigma^2\right) \tag{81}$$

$$\overset{(4)}{\leq} \frac{2}{M^2}\sum_{i=1}^{M}\left(\hat{\omega}^2\mathbb{E}[\|v_{t,i}\|^2 | x_t] + \sigma^2\right) \tag{82}$$

$$\overset{(5)}{\leq} \frac{4}{M^2}\sum_{i=1}^{M}\left(\hat{\omega}^2\mathbb{E}[\|v_{t,i} - \nabla f(x_t)\|^2 | x_t] + \hat{\omega}^2\|\nabla f(x_t)\|^2\right) + \frac{2\sigma^2}{M} \tag{83}$$

$$\overset{(6)}{\leq} \frac{2(2\hat{\omega}^2 + 1)\sigma^2}{M} + \frac{4}{M^2}\sum_{i=1}^{M}\hat{\omega}^2\|\nabla f(x_t)\|^2 \tag{84}$$

$$\overset{(7)}{\leq} \frac{2(2\hat{\omega}^2 + 1)\sigma^2}{M} + \frac{8\hat{\omega}^2 L}{M}(f(x_t) - f(x^*)) \tag{85}$$

where (1) follows since $\tilde{g}_t = \frac{1}{M}\sum_{i=1}^{M}\tilde{g}_{t,i}$ and the data samples are *i.i.d*, (2) and (5) since $\|a+b\|^2 \leq 2\|a\|^2 + 2\|b\|^2, \forall a, b \in \mathbb{R}^d$, (3) and (6) by Assumption 2.2, (4) by Eq. (3) since our MLMC compressor is unbiased (see Lemma 3.2), and (7) by Lemma F.1 since $f$ is $L$-smooth by Assumption 2.1. Now, plugging this result back into Eq. (77) yields:

$$\mathbb{E}[f(\bar{x}_T) - f(x^*)] \leq \frac{1}{T}\sum_{t=1}^{T}\mathbb{E}[f(x_t) - f(x^*)] \leq \frac{D^2}{2\eta T} + \frac{\eta}{T}\sum_{t=1}^{T}\mathbb{E}V_t^2 \tag{86}$$

$$\leq \frac{D^2}{2\eta T} + \eta\frac{2(2\hat{\omega}^2 + 1)\sigma^2}{M} + \eta\frac{8\hat{\omega}^2 L}{MT}\sum_{t=1}^{T}\mathbb{E}[f(x_t) - f(x^*)] \tag{87}$$

Choosing $\eta \leq \frac{M}{16\hat{\omega}^2 L}$ and rearranging, we have:

$$\frac{1}{T}\sum_{t=1}^{T}\mathbb{E}[f(x_t) - f(x^*)] \leq \frac{D^2}{\eta T} + \eta\frac{4(2\hat{\omega}^2 + 1)\sigma^2}{M}. \tag{88}$$

Thus, for $\eta \leq \min\left\{\frac{1}{2L}, \frac{M}{16\hat{\omega}^2 L}, \frac{D\sqrt{M}}{2\sigma\sqrt{(2\hat{\omega}^2+1)T}}\right\}$, we have:

$$\mathbb{E}[f(\bar{x}_T) - f(x^*)] \leq \frac{1}{T}\sum_{t=1}^{T}\mathbb{E}[f(x_t) - f(x^*)] \leq \frac{2D^2 L}{T} + \frac{16\hat{\omega}^2 D^2 L}{MT} + \frac{2\sigma\sqrt{2\hat{\omega}^2+1}D}{\sqrt{MT}}. \tag{89}$$

**Homogeneous Nonconvex case.**

The proof here follows very similarly to the one in the convex case. Here, similarly, we use the following bound, which holds for $\eta \leq \frac{1}{L}$ (see Appendix A.2 in (Dorfman et al., 2023)):

$$\frac{1}{T}\sum_{t=1}^{T}\mathbb{E}[\|\nabla f(x_t)\|^2] \leq \frac{2\Delta_1}{T\eta} + \frac{\eta L}{T}\sum_{t=1}^{T}\mathbb{E}V_t^2. \tag{90}$$

where $\Delta_1 = f(x_1) - f(x^*)$ and $V_t^2 = \mathbb{E}[\|\tilde{g}_t - \nabla f(x_t)\|^2 \,|x_t]$. Plugging in the expression for $V_t^2$ in Eq. (84) yields:

$$\frac{1}{T}\sum_{t=1}^{T}\mathbb{E}[\|\nabla f(x_t)\|^2] \leq \frac{2\Delta_1}{T\eta} + \eta\frac{2(2\hat{\omega}^2+1)\sigma^2 L}{M} + \eta\frac{4\hat{\omega}^2 L}{MT}\sum_{t=1}^{T}\mathbb{E}\|\nabla f(x_t)\|^2 \tag{91}$$

Choosing $\eta \leq \frac{M}{8\hat{\omega}^2 L}$ and rearranging, we have:

$$\frac{1}{T}\sum_{t=1}^{T}\mathbb{E}[\|\nabla f(x_t)\|^2] \leq \frac{4\Delta_1}{T\eta} + \eta\frac{4(2\hat{\omega}^2+1)\sigma^2 L}{M}. \tag{92}$$

Thus, for $\eta \leq \min\left\{\frac{1}{L}, \frac{M}{8\hat{\omega}^2 L}, \frac{\sqrt{M}}{\sigma\sqrt{(2\hat{\omega}^2+1)LT}}\right\}$, we have:

$$\frac{1}{T}\sum_{t=1}^{T}\mathbb{E}[\|\nabla f(x_t)\|^2] \leq \frac{4\Delta_1 L}{T} + \frac{32\hat{\omega}^2\Delta_1 L}{MT} + \frac{4\sigma\sqrt{(2\hat{\omega}^2+1)L}}{\sqrt{MT}}. \tag{93}$$

$\square$

### F.2. Self-bounding Property of Smooth Functions

**Lemma F.1.** *A function $f : \mathbb{R}^d \to \mathbb{R}$ that is $L$-smooth (see Assumption 2.1) satisfies the following, for any $x \in \mathbb{R}^d$:*

$$\|\nabla f(x)\|^2 \leq 2L(f(x) - f(x^*)), \tag{94}$$

*where $x^* \in \arg\min_x f(x)$.*

*Proof.* Note that $f(x^*) \leq f(x')$, for any $x' \in \mathbb{R}^d$, by definition of $x^*$. Thus we have, for $x' = x - \frac{1}{L}\nabla f(x)$:

$$f(x^*) \leq f\left(x - \frac{1}{L}\nabla f(x)\right) \tag{95}$$

$$\overset{(1)}{\leq} f(x) - \frac{1}{L}\|\nabla f(x)\|^2 + \frac{L}{2}\frac{1}{L^2}\|\nabla f(x)\|^2, \tag{96}$$

where (1) follows by the smoothness of $f$. Rearranging yields:

$$\|\nabla f(x)\|^2 \leq 2L(f(x) - f(x^*)). \tag{97}$$

$\square$

### F.3. Parallelization Guarantees

Our method, formalized in Alg. 2 (nonadaptive) and Alg. 3 (adaptive) produces unbiased gradient estimates. Therefore, the error bound of our method is very similar to that of Alg. 1 (data-parallel SGD). concretely, Alg. 2-3 guarantee the following error bounds in the convex and nonconvex cases, and in the homogeneous setting, respectively (Theorem 4.1):

$$\mathbb{E}[f(\bar{x}_T) - f(x^*)] \in \mathcal{O}\left( \frac{D^2 L}{T} + \frac{\hat{\omega}^2 D^2 L}{MT} + \frac{(\hat{\omega}+1)\sigma D}{\sqrt{MT}} \right) \tag{98}$$

$$\frac{1}{T}\sum_{t=1}^{T} \mathbb{E}\|\nabla f(x_t)\|^2 \in \mathcal{O}\left( \frac{\Delta_1 L}{T} + \frac{\hat{\omega}^2 \Delta_1 L}{MT} + \frac{(\hat{\omega}+1)\sigma\sqrt{L}}{\sqrt{MT}} \right) \tag{99}$$

Note that the middle terms are asymptotically negligible, and therefore these bounds are asymptotically similar to the bounds guaranteed by Alg. 1, i.e.:

$$\mathcal{O}\left( \frac{1}{T} + \frac{\sigma}{\sqrt{MT}} \right), \tag{100}$$

albeit with the an increased variance $(\hat{\omega}^2 + 1)\sigma$ (that depends on the baseline compression method we use, e.g., top-$k$ or fixed-point compression) instead of $\sigma$. Note that $\hat{\omega}$ depends on the compressor and therefore on the compression coefficient $\alpha$. Please refer to Appendices B, D, E, for exact calculations for certain examples. Note that the same *asymptotic* bound holds for the heterogeneous case, with the heterogeneity bound $\xi$ added to $\sigma$.

In contrast, biased compression methods utilize an error correction mechanism to account for the bias. While these achieve impressive results, additional terms are added to the error bounds due to the bias of the gradients). Specifically, EF21-SGDM (Fatkhullin et al., 2023) guarantees the following bound (Corollary 3 in Fatkhullin et al. (2023), nonconvex case):

$$\frac{1}{T}\sum_{t=1}^{T} \mathbb{E}\|\nabla f(x_t)\|^2 \in \mathcal{O}\left( \frac{\Delta_1 L}{\alpha T} + \frac{\Delta_1 L \sigma^{1/2}}{\alpha^{1/2} T^{3/4}} + \frac{\Delta_1 L \sigma}{\sqrt{MT}} \right) \tag{101}$$

Let us consider our bounds in Eq. (98)-(99). Note that asymptotically, the third term is dominant. Therefore, $M$ can be as large as $o(T)$, or equivalently $o(\sqrt{N})$, where $N$ is the size of the whole dataset, without a degradation in performance. Moreover, these bounds can be written in terms of the size of the dataset, $N$, since $T = N/M$ ($N$ points split on $M$ machines). Thus, asymptotically, performance starts to degrade when (neglecting constants other than $T$ and $M$ (which depends on $T$)):

$$\frac{1}{\sqrt{MT}} \geq \frac{1}{T} \iff M \leq T \iff M \leq \sqrt{N}. \tag{102}$$

Now, similarly considering the parallelization limit of the bound of EF21-SGDM in Eq. (101), and note that the second term is always more dominant than the first, asymptotic performance starts to degrade when:

$$\frac{1}{\sqrt{MT}} \geq \frac{1}{T^{3/4}} \iff M \leq \sqrt{T} \iff M \leq N^{1/3}, \tag{103}$$

which implies a parallelization limit of up to $o(\sqrt{T})$, or equivalently $o(N^{1/3})$, without a degradation in performance. This shows that in the regime of *massive* parallelization, i.e., when $M$ is very large, our method enables better (more) parallelization without a degradation in performance. Interestingly, when $M$ is small, EF21-SGDM might have a slight edge, as the dominant terms are $\frac{\sigma}{\sqrt{MT}}$ for EF21-SGDM and $\frac{(\hat{\omega}^2+1)\sigma}{\sqrt{MT}}$ for our method, since MLMC methods tranduce bias into variance, increasing it slightly. However, our experimental results show that our method maintains its edge over EF21-SGDM even when $M$ is very small.

### F.4. Extension to the Heterogeneous Case

Our method can be naturally extended to the heterogeneous case in which each machine samples points from a different distribution. That is, each machine $i \in [M]$ can sample $i.i.d$ data from some data distribution $\mathcal{D}_i$. We assume that the heterogeneity is bounded, namely there exists $\xi \geq 0$ such that, $\forall x \in \mathbb{R}^d$:

$$\frac{1}{M}\sum_{i=1}^{M} \|\nabla f_i(x) - \nabla f(x)\|^2 \leq \xi^2 \tag{104}$$

Under this assumption, our method guarantees the bounds formalized in Theorem F.2 in the convex and nonconvex cases.

**Theorem F.2.** *Under Assumptions 2.1-2.2, and the bounded heterogeneity assumption in Eq. (104), Alg. 2 and Alg. 3 guarantee the following error bounds in the heterogeneous convex and nonconvex cases, respectively:*

$$\mathbb{E}[f(\bar{x}_T) - f(x^*)] \in \mathcal{O}\left(\frac{D^2 L}{T} + \frac{\hat{\omega}^2 D^2 L}{MT} + \frac{\hat{\omega}(\sigma + \xi)D}{\sqrt{MT}} + \frac{\sigma D}{\sqrt{MT}}\right)$$

$$\frac{1}{T}\sum_{t=1}^{T} \mathbb{E}\|\nabla f(x_t)\|^2 \in \mathcal{O}\left(\frac{\Delta_1 L}{T} + \frac{\hat{\omega}^2 \Delta_1 L}{MT} + \frac{\hat{\omega}(\sigma + \xi)\sqrt{L}}{\sqrt{MT}} + \frac{\sigma\sqrt{L}}{\sqrt{MT}}\right)$$

*Proof.* **Heterogeneous Convex Case.**
Similarly to the homogeneous case, since the MLMC gradients, $\tilde{g}_{t,i}$ used in Alg. 2-3 are unbiased estimators of the true machine-specific gradients, namely $\mathbb{E}[\tilde{g}_{t,i}|x_t] = \nabla f_i(x_t), \forall t \in [T], \forall i \in [M]$, we have the following bound in the convex case, for $\eta \leq \frac{1}{2L}$ ((Dorfman et al., 2024)):

$$\mathbb{E}[f(\bar{x}_T) - f(x^*)] \leq \frac{1}{T}\sum_{t=1}^{T}\mathbb{E}[f(x_t) - f(x^*)] \leq \frac{D^2}{2\eta T} + \frac{\eta}{T}\sum_{t=1}^{T}\mathbb{E}V_t^2 \tag{105}$$

where $\bar{x}_T = \frac{1}{T}\sum_{t=1}^{T}$, $x^* = \arg\min_x f(x)$, $D = \|x_1 - x^*\|$, and $V_t^2 = \mathbb{E}[\|\tilde{g}_t - \nabla f(x_t)\|^2 |x_t]$. We now consider the term $V_t^2$:

$$V_t^2 = \mathbb{E}[\|\tilde{g}_t - \nabla f(x_t)\|^2 |x_t] \tag{106}$$

$$= \mathbb{E}\left[\left\|\frac{1}{M}\sum_{i=1}^{M}(\tilde{g}_{t,i} - \nabla f_i(x_t))\right\|^2 \Big| x_t\right] \tag{107}$$

$$\stackrel{(1)}{=} \frac{1}{M^2}\sum_{i=1}^{M}\mathbb{E}[\|\tilde{g}_{t,i} - \nabla f_i(x_t)\|^2 |x_t] \tag{108}$$

$$\stackrel{(2)}{\leq} \frac{2}{M^2}\sum_{i=1}^{M}\left(\mathbb{E}[\|\tilde{g}_{t,i} - v_{t,i}\|^2 |x_t] + \mathbb{E}[\|v_{t,i} - \nabla f_i(x_t)\|^2 |x_t]\right) \tag{109}$$

$$\stackrel{(3)}{\leq} \frac{2}{M^2}\sum_{i=1}^{M}\left(\mathbb{E}[\|\tilde{g}_{t,i} - v_{t,i}\|^2 |x_t] + \sigma^2\right) \tag{110}$$

$$\stackrel{(4)}{\leq} \frac{2}{M^2}\sum_{i=1}^{M}\left(\hat{\omega}^2\mathbb{E}[\|v_{t,i}\|^2 |x_t] + \sigma^2\right) \tag{111}$$

$$\stackrel{(5)}{\leq} \frac{6\hat{\omega}^2}{M^2}\sum_{i=1}^{M}\left(\mathbb{E}[\|v_{t,i} - \nabla f_i(x_t)\|^2 |x_t] + \|\nabla f_i(x_t) - \nabla f(x_t)\|^2 + \|\nabla f(x_t)\|^2\right) + \frac{2\sigma^2}{M} \tag{112}$$

$$\stackrel{(6)}{\leq} \frac{2(3\hat{\omega}^2 + 1)\sigma^2}{M} + \frac{6\hat{\omega}^2\xi^2}{M} + \frac{6\hat{\omega}^2}{M^2}\sum_{i=1}^{M}\|\nabla f(x_t)\|^2 \tag{113}$$

$$\stackrel{(7)}{\leq} \frac{2(3\hat{\omega}^2 + 1)\sigma^2}{M} + \frac{6\hat{\omega}^2\xi^2}{M} + \frac{12\hat{\omega}^2 L}{M}(f(x_t) - f(x^*)) \tag{114}$$

where (1) follows since $\tilde{g}_t = \frac{1}{M}\sum_{i=1}^{M}\tilde{g}_{t,i}$ and the data samples are $i.i.d$, (2) since $\|a + b\|^2 \leq 2\|a\|^2 + 2\|b\|^2, \forall a, b \in \mathbb{R}^d$, (3) by Assumption 2.2, (4) by Eq. (3) since our MLMC compressor is unbiased (see Lemma 3.2), (5) since $\|a + b + c\|^2 \leq 3(\|a\|^2 + \|b\|^2 + \|c\|^2), \forall a, b, c \in \mathbb{R}^d$, (6) by Assumptions 2.2 and Eq. (104), and (7) by Lemma F.1 since $f$ is $L$-smooth by

Assumption 2.1. plugging this result back into Eq. (105) yields:

$$\mathbb{E}[f(\bar{x}_T) - f(x^*)] \leq \frac{1}{T}\sum_{t=1}^{T}\mathbb{E}[f(x_t) - f(x^*)] \leq \frac{D^2}{2\eta T} + \frac{\eta}{T}\sum_{t=1}^{T}\mathbb{E}V_t^2 \tag{115}$$

$$\leq \frac{D^2}{2\eta T} + \eta\frac{2(3\hat{\omega}^2 + 1)\sigma^2}{M} + \eta\frac{6\hat{\omega}^2\xi^2}{M} + \eta\frac{12\hat{\omega}^2 L}{MT}\sum_{t=1}^{T}\mathbb{E}[f(x_t) - f(x^*)] \tag{116}$$

Choosing $\eta \leq \frac{M}{24\hat{\omega}^2 L}$ and rearranging, we have:

$$\frac{1}{T}\sum_{t=1}^{T}\mathbb{E}[f(x_t) - f(x^*)] \leq \frac{D^2}{\eta T} + \eta\frac{4(3\hat{\omega}^2 + 1)\sigma^2 + 12\hat{\omega}^2\xi^2}{M}. \tag{117}$$

Thus, for $\eta \leq \min\left\{\frac{1}{2L}, \frac{M}{16\hat{\omega}^2 L}, \frac{D\sqrt{M}}{\sqrt{4(3\hat{\omega}^2+1)\sigma^2 + 12\hat{\omega}^2\xi^2 T}}\right\}$, we have:

$$\mathbb{E}[f(\bar{x}_T) - f(x^*)] \leq \frac{1}{T}\sum_{t=1}^{T}\mathbb{E}[f(x_t) - f(x^*)] \leq \frac{2D^2 L}{T} + \frac{16\hat{\omega}^2 D^2 L}{MT} + \frac{\sqrt{4(3\hat{\omega}^2 + 1)\sigma^2 + 12\hat{\omega}^2\xi^2}D}{\sqrt{MT}} \tag{118}$$

Therefore:

$$\mathbb{E}[f(\bar{x}_T) - f(x^*)] \in \mathcal{O}\left(\frac{D^2 L}{T} + \frac{\hat{\omega}^2 D^2 L}{MT} + \frac{\hat{\omega}(\sigma + \xi)D}{\sqrt{MT}} + \frac{\sigma D}{\sqrt{MT}}\right). \tag{119}$$

Note that this bound is consistent with its homogeneous counterpart when $\xi = 0$.

**Heterogeneous Nonconvex Case.**

The proof follows very similarly to the one for the convex case. Here, we have for $\eta \leq \frac{1}{L}$ ((Dorfman et al., 2024)):

$$\frac{1}{T}\sum_{t=1}^{T}\mathbb{E}[\|\nabla f(x_t)\|^2] \leq \frac{2\Delta_1}{T\eta} + \frac{\eta L}{T}\sum_{t=1}^{T}\mathbb{E}V_t^2. \tag{120}$$

where $\Delta_1 = f(x_1) - f(x^*)$ and $V_t^2 = \mathbb{E}[\|\tilde{g}_t - \nabla f(x_t)\|^2 \,|x_t]$. Plugging in the expression for $V_t^2$ in Eq. (113) yields:

$$\frac{1}{T}\sum_{t=1}^{T}\mathbb{E}[\|\nabla f(x_t)\|^2] \leq \frac{2\Delta_1}{T\eta} + \eta\frac{2(3\hat{\omega}^2 + 1)\sigma^2 L}{M} + \eta\frac{6\hat{\omega}^2\xi^2 L}{M} + \eta\frac{6\hat{\omega}^2 L}{MT}\sum_{t=1}^{T}\mathbb{E}\|\nabla f(x_t)\|^2 \tag{121}$$

Choosing $\eta \leq \frac{M}{12\hat{\omega}^2 L}$ and rearranging, we have:

$$\frac{1}{T}\sum_{t=1}^{T}\mathbb{E}[\|\nabla f(x_t)\|^2] \leq \frac{4\Delta_1}{T\eta} + \eta\frac{(4(3\hat{\omega}^2 + 1)\sigma^2 + 12\hat{\omega}^2\xi^2)L}{M}. \tag{122}$$

Thus, for $\eta \leq \min\left\{\frac{1}{L}, \frac{M}{12\hat{\omega}^2 L}, \frac{\sqrt{M}}{\sqrt{((3\hat{\omega}^2+1)\sigma^2 + 3\hat{\omega}^2\xi^2)LT}}\right\}$, we have:

$$\frac{1}{T}\sum_{t=1}^{T}\mathbb{E}[\|\nabla f(x_t)\|^2] \leq \frac{4\Delta_1 L}{T} + \frac{48\hat{\omega}^2\Delta_1 L}{MT} + \frac{4\sqrt{((3\hat{\omega}^2 + 1)\sigma^2 + 3\hat{\omega}^2\xi^2)L}}{\sqrt{MT}}. \tag{123}$$

Therefore:

$$\frac{1}{T}\sum_{t=1}^{T}\mathbb{E}[\|\nabla f(x_t)\|^2] \in \mathcal{O}\left(\frac{\Delta_1 L}{T} + \frac{\hat{\omega}^2\Delta_1 L}{MT} + \frac{\hat{\omega}(\sigma + \xi)\sqrt{L}}{\sqrt{MT}} + \frac{\sigma\sqrt{L}}{\sqrt{MT}}\right) \tag{124}$$

Note that his bound is consistent with its homogeneous counterpart when $\xi = 0$. $\qquad\square$

# G. Additional Experiments

### G.1. Sparsification Compressors Evaluation on CIFAR-10 Image Classification Using ResNet18

We ran additional experiments comparing the performance of our MLMC-Top-$k$ compression method (Alg. 3), Top-$k$, Rand-$k$, EF21-SGDM, and uncompressed SGD, on CIFAR-10 Image Classification using ResNet18.

Figure 4 shows the test accuracy of the compared algorithms as a function of communication complexity (the number of transmitted bits), for $M = 4$ machines and a batch size of 128 (top quartet) and $M = 32$ machines and a batch size of 64 (bottom quartet), and for various levels of sparsification $k \in \{0.001, 0.005, 0.01, 0.05\}n$, where $n \approx 1.1 \times 10^7$ is the number of model parameters. The results were averaged over 5 different seeds to mitigate randomness. We display these results as a function of the epoch (the number of iterations) in Figure 5.

Figures 4-5 show that our method demonstrates an advantage over the others in terms of convergence speed and test accuracy.

### G.2. RTN Compression Evaluation on BERT Finetuning on GLUE SST-2

We evaluate the performance of our MLMC-compression scheme on quantization-based compressors. Specifically we consider Round-to-Nearest (RTN) compression. Given a vector $v$, RTN compresses $v$ by defining a quantization grid and rounding each element of $v$ to the nearest integer on this grid. The spacing of this grid is controlled by the quantization step-size, $\delta^l$, where $l$ defines the quantization level (a larger $l$ corresponds to finer quantization). Specifically, given a vector $v$, its RTN-compression (of level $l$) is given by

$$C_{RTN}^l(v) = \delta^l \cdot \text{clip}(\text{round}(v/\delta^l), -c, c), \tag{125}$$

where the division, rounding, and clipping are done in an element-wise manner, and "round" rounds each element to its nearest integer on the grid defined by $\delta^l = \frac{2c}{2^l - 1}$.

We evaluated our Adaptive MLMC-compression method (Alg. 3) with the RTN compressor as a baseline (which we term MLMC-RTN), and compared it to regular RTN compression (without MLMC) with $l \in \{2, 4, 8, 16\}$, and to uncompressed SGD, for $M = 4$ and $M = 32$ machines. We used a batch size of 16 and averaged over 5 different seeds to mitigate randomness. We present the results in Figure 6. Note that our method enjoys a significant advantage in communication efficiency compared to the others in this case as well. Interestingly, the performance of all methods in terms of iteration efficiency is very similar, with SGD having a slight advantage over the others.

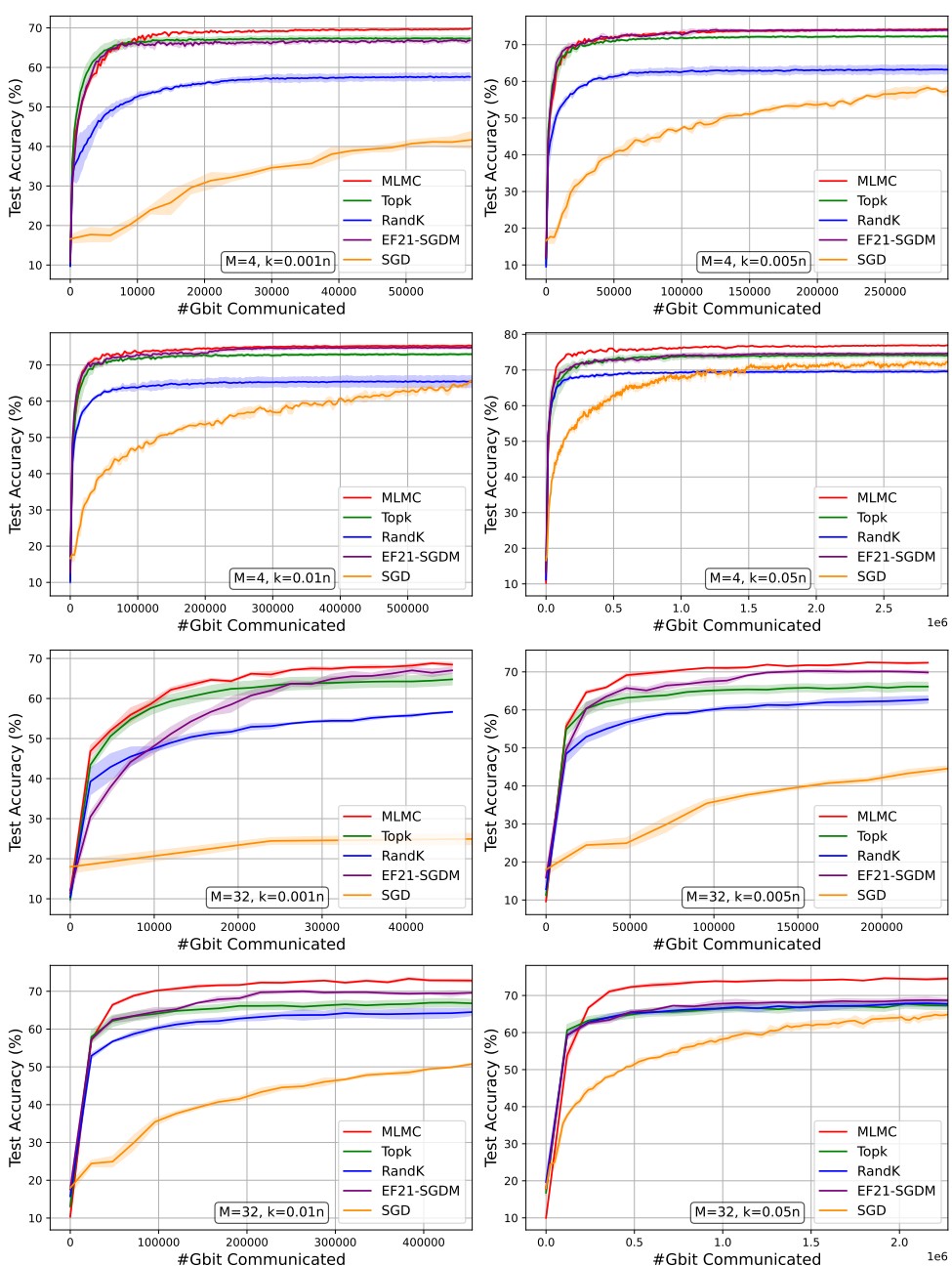

*Figure 4.* CIFAR-10 image classification using ResNet18, **communication efficiency** comparison of our MLMC-Top-$k$ Compressor (Alg. 3),Top-$k$, Rand-$k$, EF21-SGDM, and uncompressed SGD, for $M = 4$ machines and a batch size of $128$ and for $M = 32$ machines and a batch size of $64$, averaged over 5 different seeds.

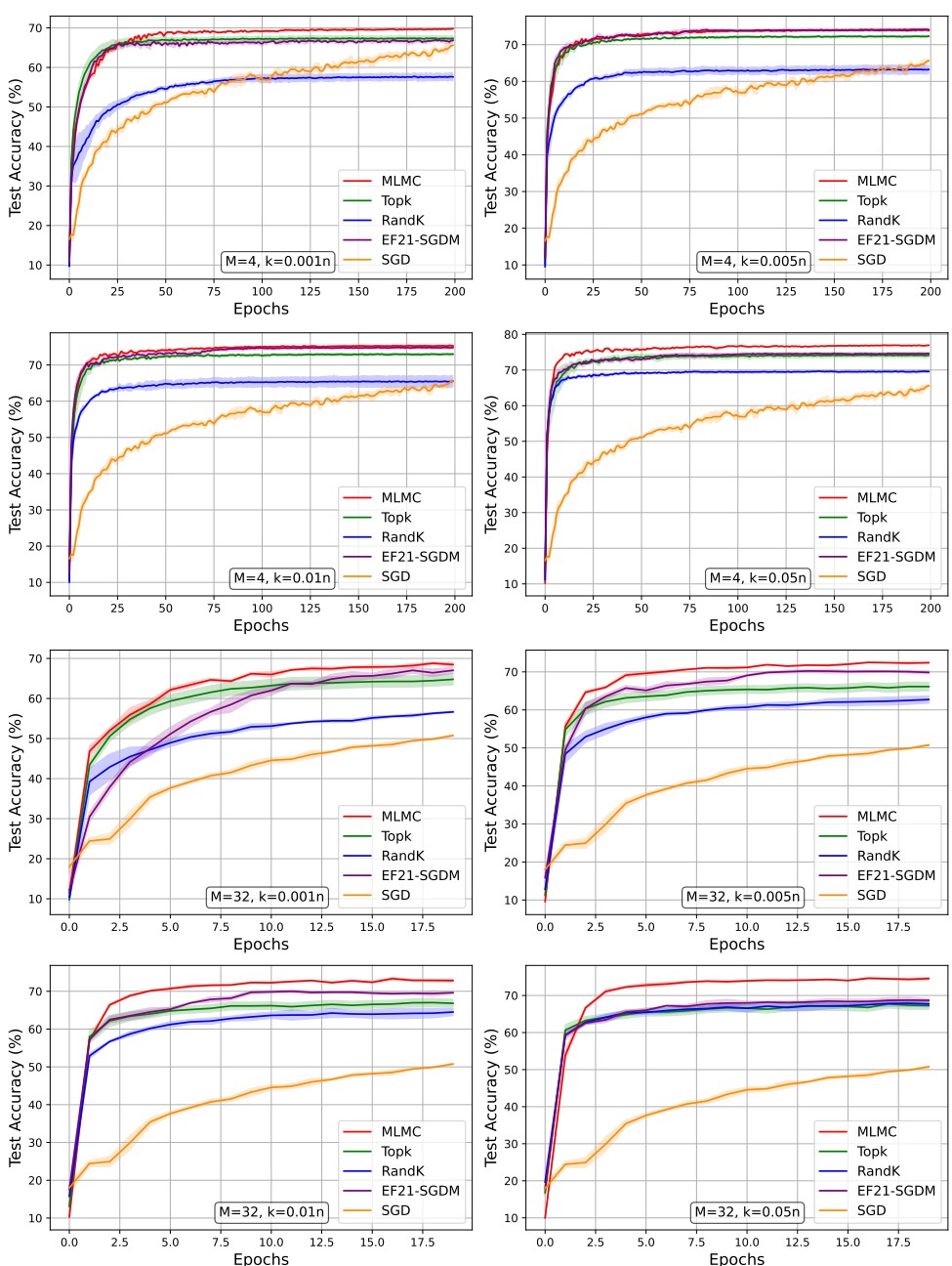

*Figure 5.* CIFAR-10 image classification using ResNet18, **iteration efficiency** comparison of our MLMC-Top-$k$ Compressor (Alg. 3),Top-$k$, Rand-$k$, EF21-SGDM, and uncompressed SGD, for $M = 4$ machines and a batch size of 128 and for $M = 32$ machines and a batch size of 64, averaged over 5 different seeds.

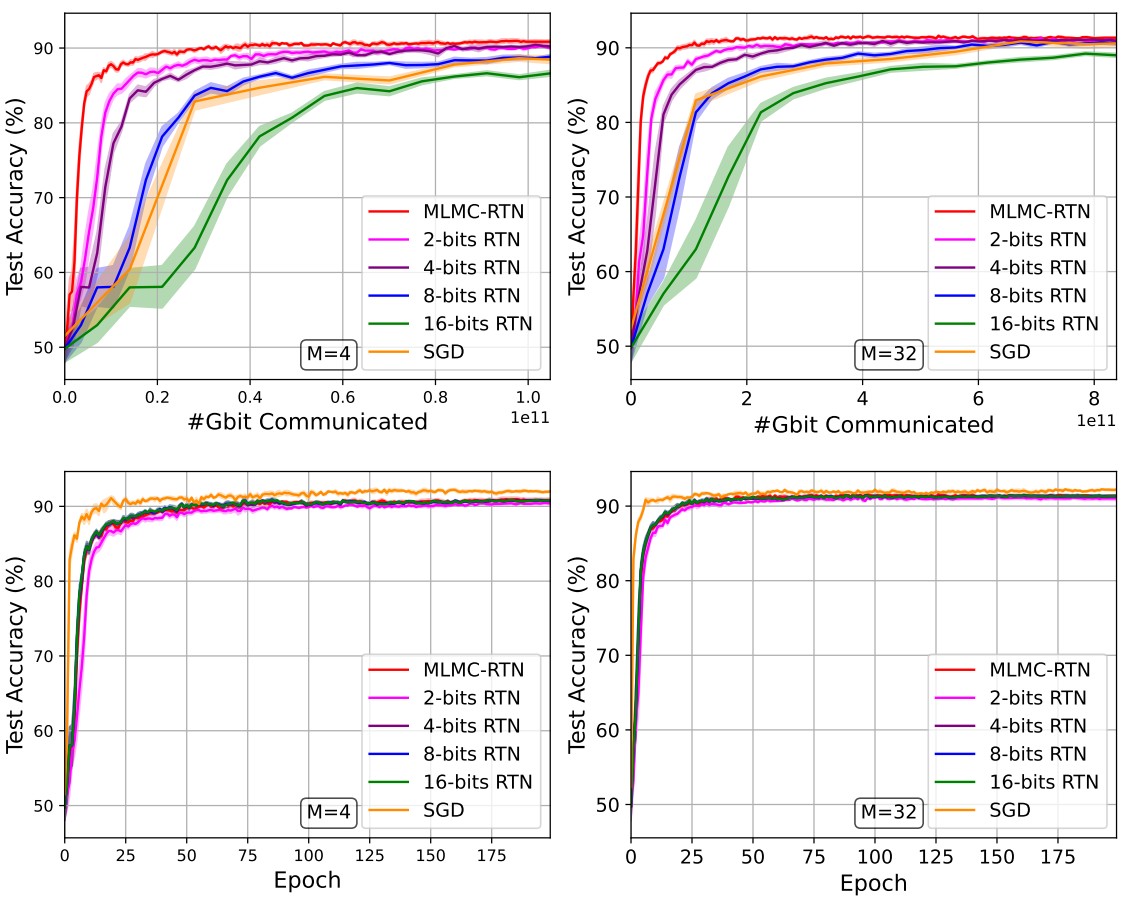

*Figure 6.* Finetuning BERT on GLUE SST2 **communication efficiency** (top row) and **iteration efficiency** (bottom row) comparison of the Adaptive MLMC-RTN (Alg. 3), RTN with $l \in \{2, 4, 8, 16\}$, and uncompressed SGD, for $M = 4$ and $M = 32$ machines and a batch size of 16 samples. The results are averaged over 5 different seeds.

