# OpenReview forum: "Beyond Communication Overhead: A Multilevel Monte Carlo Approach for Mitigating Compression Bias in Distributed Learning"
_ICML.cc/2025/Conference — ICML 2025 poster_

### Official Review · Reviewer_Fo3R · 2025-03-12

**Overall Recommendation:** 3

**Summary:**

The paper introduces a Multilevel Monte Carlo compression scheme that leverages biased compressors to construct unbiased gradient estimates. The proposed approach aims to combine the empirical efficiency of biased compressors (Top-k, bitwise compression) with the theoretical guarantees of unbiased methods.

**Claims And Evidence:**

yes

**Essential References Not Discussed:**

none

**Experimental Designs Or Analyses:**

yes

**Methods And Evaluation Criteria:**

yes

**Other Comments Or Suggestions:**

none

**Other Strengths And Weaknesses:**

Strength:
1. The paper is well-written and easy to follow.
2. The paper introduces an innovative way to bridge biased and unbiased compression techniques.
3. The authors provide thorough theoretical analysis of their method, including detailed proofs of unbiasedness and variance bounds.

Weakness:
3. The paper does not provide explicit per-iteration time cost comparisons.

**Questions For Authors:**

1. As acknowledged in the paper, the MLMC approach trades bias for increased variance, which might impact performance in some scenarios, such as setup with a small number of machines. Is this the reason when with 4 machines, the performance gain is not that huge?
3. Can the author provide per iteration time cost for different schemes?
3. Is MLMC in Figure 3 based on Algorithm 2 or 3?

**Relation To Broader Scientific Literature:**

satisfactory

**Theoretical Claims:**

yes

---

> ### Author Rebuttal · Authors · 2025-04-01
>
> We thank the reviewer for their positive evaluation of our paper and for the constructive feedback. Below, we address the questions raised:
>
> **1. Scaling with number of machines**
>
> You are right that the performance gains from MLMC grow with the level of parallelization. When using only 4 machines, variance reduction is not as effective, so the gains are naturally smaller. However, as shown in our scalability plots (e.g., Figure 1 ($M=4$) compared to Figure 2 ($M=32$)), MLMC achieves significant speedups, smaller errors, and improved communication efficiency in large-scale distributed settings.
>
> This is a key strength of our method. It retains the empirical efficiency of biased compressors while providing the unbiasedness and theoretical guarantees needed for scalable, stable learning in high-parallelism regimes. Moreover, higher parallelization induces a better variance reduction effect, which makes our MLMC estimator even better.
>
> We will clarify this in the revised paper to better highlight the scaling advantages of MLMC.
>
> **2. Computational overhead**
>
> That is a good point. The computational overhead of MLMC is comparable to standard methods such as Top-k and AdaGrad, and is often negligible relative to the overall training time. Similar works also focus less on the computational overhead since it is often negligible compared to the overhead introduced by the communication.
>
> Specifically, using top-$k$, for example, incurs $O(d\log(k))$ computational complexity per iteration and per machine while our adaptive MLMC method (Alg. 3) incurs $O(d\log(d))$, which is a very small difference in practical scenarios.
>
> In more detail, using top-$k$ requires finding the $k$ largest elements, which costs $O(d\cdot\log(k))$ in each iteration and for each machine.
> In contrast, our adaptive MLMC method (Alg. 3) with top-$k$ requires sorting the vector first, costing $O(d\cdot\log(d))$, and computing the probabilities and constructing the MLMC estimator, which costs $O(d)$ in total (for computing the norm of the vector, similar to AdaGrad, and for picking the $l$-th largest element, which costs $O(1)$ since the vector is sorted).
> However, you are right that this is worth discussing to improve clarity, and we will add it to our paper.
>
> **3. Clarification on Figure 3**
>
> Thank you for pointing this out. The results in Figure 3 correspond to Algorithm 2. We will clarify this in the figure caption and text to avoid confusion.
>
> **4. Experiments**
>
> We ran new experiments, including NLP experiments using BERT on SST-2. These are anonymously available at "https://anonymous.4open.science/r/ICML2025MLMC-5346".
>
> Thank you again for your constructive feedback!

---

> > ### Comment · Reviewer_Fo3R · 2025-04-08
> >
> > I thank the authors for the response and providing the new comments. I like how the paper  combines the empirical efficiency of biased compressors with theoretical guarantees of unbiased ones (and goes beyond importance sampling). The response clarified my concerns. I will maintain my score.

---

> > > ### Author Response · Authors · 2025-04-09
> > >
> > > Dear reviewer,
> > >
> > > We thank you sincerely for your comment and for acknowledging the novelty and contribution of our paper beyond importance sampling.

---

### Official Review · Reviewer_yf8R · 2025-03-12

**Overall Recommendation:** 2

**Summary:**

The work proposed to consider Multilevel Monte Carlo (MLMC) in Distributed Learning to mitigate the problem with unbiased compressors analysis. Work introduced a novel Multilevel Monte Carlo (MLMC) compression scheme that leverages biased compressors to construct statistically unbiased estimates.

**Claims And Evidence:**

The paper in general is written pretty well.

**Major:**

1. Unfortunately, I have concerns about the paper. The authors highlighted the possibility of using an interesting mechanism. However, the proof presented in Parallelization 3.4, the second paragraph, is not complete. The MLMC is unbiased and satisfies the Assumption hidden in Line 117, but nothing has been analyzed in terms of variance in terms of Assumption 2.2. For me, the proof is a bit artificial and the authors should elaborate way more on the Analysis.  Essentially, I don't like the fact that the original estimator is replaced by another and nothing has been said in terms of its variance.

2. I'm pretty skeptical that if use L=2, and use for L=2 identical mapping and for L=1 TopK[k=1] compressor the \alpa=k/d does not come into the rate (2) at all. However, the authors claim: "Note that since our MLMC gradient estimates are unbiased, a similar error bound to Eq. (2) holds"

3. It's great that you have auxiliary Lemmas, but please formulate and prove the convergence theorem in detail (either for convex or non-convex case).

**Minor:**

4. Please use the notation in Line 135 to highlight the estimator of \nabla f_i. You notation is slightly overload because you have both f_i(x) and f_i(x,z)

5. Please elaborate more on the fact that rate (2) is optimal for convex setting in terms of rate for Stochastic  Gradient Descend.

6. Please, if Assumption 3.2 is too restrictive, use "Ahmed Khaled, Othmane Sebbouh, Nicolas Loizou, Robert M. Gower, and Peter Richtarik, Unified Analysis of Stochastic Gradient Methods for Composite Convex and Smooth Optimization, arXiv:2006.11573, 2020"

7. Please elaborate more on the rate for non-convex cases. The rates 1/T to 1/sqrt{T} depend on assumptions and methods.

8.  > This way, although the compressed gradients can be biased, their MLMC estimators are always unbiased.

I can take take expectation of (5) and get line 187 right column. But the estimator is biased or not will -- depend on X^L.
(please rephrase)

9. Please be more concrete in Definition 3.1. and specify that C^i, i \in [L], i \ne L can be any compressor (1) or (2).

10. Please add experiments with Convex Setting (Quadratics or Logistic Regression) by selecting step-size according to theory to demonstrate the correctness of your method.

**Essential References Not Discussed:**

All related works are properly introduced and utilized.

**Ethical Review Concerns:**

None.

**Experimental Designs Or Analyses:**

Yes. In the Applied Optimization sense, methods sound good.

**Methods And Evaluation Criteria:**

In Applied Optimization sense, there are experiments wth training ResNet-18.
In a more restrictive setting with trying to eliminate humans from the loop during training, there are no such experiments.

**Other Comments Or Suggestions:**

No.

**Other Strengths And Weaknesses:**

The paper is well written, but requires more elaborated work on Theory and minor polishing.

**Questions For Authors:**

See Claims And Evidence section.

**Relation To Broader Scientific Literature:**

The proposed methodology is interesting and has serious potential to revolutionize how we think about ways to mitigate problems with the analysis of biased compressors.

**Theoretical Claims:**

Appendix A, D.

---

> ### Author Rebuttal · Authors · 2025-04-01
>
> We thank the reviewer for the thoughtful and encouraging feedback, and for pointing out areas where the theoretical analysis could be clarified. We respond to each concern below and will revise the paper accordingly.
>
> **1. Variance Analysis and Convergence Guarantee**
>
> We appreciate your observation regarding the variance and convergence analysis.
>
> Regarding the variance, we presented the full derivations of the variance of the MLMC estimators as part of the calculation of the optimal probability distribution (which is optimized to minimize the variance) in Appendices B,C, and D (see Eq. (33, 44, 55)).
>
> For a more practical example, in Lemma 3.6 we show that the variance of our MLMC estimator in the exponential distribution case (see Assumption 3.5) is given by $O(1/rs)$, where $r$ is the exponential decay rate of the vector's elements, which is better than the $O(d/s)$ variance of rand-$k$ when $1/r < d$ (i.e., when the decay rate is sufficient, which is the more interesting scenario).
> Note that when the vector is nearly uniform, we have $1/r \approx d$ and the variances will be comparable in this case, as expected.
>
> Regarding convergence, please note that since our MLMC gradient estimators are unbiased by construction, convergence follows by the standard SGD convergence analysis, where the only difference is the additional variance introduced by the compression, as we state in lines 186-200. However, we agree that making these derivations explicit would strengthen the presentation, and we will add this to the paper as you suggested to make it clearer.
>
> Specifically, the formal convergence theorem for Alg. 2 and Alg. 3 will hold under Assumptions 2.1 (smoothness) and 2.2 (bounded variance) and will guarantee similar error bounds (for the convex and nonconvex cases) as the ones in Theorem 2.1 and Eq. (2), only with $\sigma_{comp}+\sigma$ in place of $\sigma$, where $\sigma_{comp}$ depends on the compressor and thus on $\alpha^l, l\in[L]$ (see e.g. Eq. (60) in Appendix D).
>
> We formalize the convergence theorem as follows.
>
> Theorem (convex case). Under Assumptions 2.1 (smoothness) and 2.2 (bounded variance), Alg. 2 (nonadaptive MLMC compression) guarantees the following error bound: $O(\frac{1}{T}+\frac{\sigma_{comp}+\sigma}{\sqrt{MT}})$
>
> Although the proof follows very similarly to the SGD convergence theorem, we will formalize and add it to the paper for completeness. Similar Theorem and proof follow for the nonconvex case and we will add them as well.
>
> **2. Bounds dependence on compression constants**
>
> We believe the reviewer’s concern refers to the apparent disappearance of level-specific compression constants (i.e., $\alpha_{t,i}^l$) in the final rate. This is a good point. As we mentioned in the previous point, these constants come to light in the variance introduced by compression, i.e., in $\sigma_{comp}$ (see e.g. Eq. (60) in Appendix D) which appears in the final convergence rate. We agree with the reviewer's suggestion to add this to the main paper to make it clearer, and we will incorporate this in the final version.
>
> **3. Minor Points**
>
> We thank the reviewer for the helpful suggestions regarding notation, definitions, and related clarity issues.
> We will revise the manuscript accordingly to improve clarity.
>
> * You are correct that the unbiasedness of our MLMC estimator depends on the highest level, $L$. Since we define $C^L(v):=v$, i.e., there is no compression, our MLMC method produces unbiased estimates of the true gradient.
>
> * Regarding the convergence rate of convex and non-convex SGD, we formalized the assumptions in our paper, but we will elaborate more on this and on the optimality of the bounds to improve clarity as you suggested.
>
> * Regarding eliminating humans from the loop in experiments, we ran additional NLP experiments using the AdamW optimizer, which employs an adaptive learning rate and alleviates the need for extensive learning rate tuning. The results are anonymously available at "https://anonymous.4open.science/r/ICML2025MLMC-5346".
>
> Thank you again for your constructive feedback!

---

### Official Review · Reviewer_LHZE · 2025-03-13

**Overall Recommendation:** 2

**Summary:**

The article presents a new compression method that uses the MLMC algorithm to turn biased compressors into unbiased ones.

**Claims And Evidence:**

The claims in the paper are correct and verified.

**Essential References Not Discussed:**

All references necessary for understanding the article are provided.

**Experimental Designs Or Analyses:**

1) Basic ResNet18+CIFAR10 training gives more than 90 percent on test. Such experiments do not make sense, we have lost 20% of quality for all operators. It seems that if we use less aggressive compression, MLMC will lose to TopK.

2) Please add to the comparison operators that compute the important sampling of coordinates.

3) Still, ResNet+CIFAR10 although a classic production, are outdated. It would be interesting to see heavier tasks that require real distributed computation (resnet on cifar on a laptop learns in a few hours). I recommend for example BERT learning or Llama finetuning.

**Methods And Evaluation Criteria:**

The proposed methods are proved under generally accepted assumptions on the target function. The algorithms are validated on the ResNEet+ Cifar10 problem, which is common.

**Other Comments Or Suggestions:**

-

**Other Strengths And Weaknesses:**

I can't recommend for acceptance just yet.

**Questions For Authors:**

-

**Relation To Broader Scientific Literature:**

The authors propose a new way of compression. It doesn't make a big breakthrough either way. Moreover, the applicability of these approaches and the big difference from the existing ones is questionable.

**Theoretical Claims:**

The proofs and facts appear to be correct.  But I'm left unclear about one thing that seems like it should definitely be clarified, how do these approaches differ from importance sampling (see Sec 2.2 from Beznosikov et al)? It looks like, all these compressors can be reduced to a simpler form. Let me provide examples:

__Bit-wise compressors:__ Here, the difference $g^l - g^{l-1}$ is used, and from the proposed compressor, it follows that $C(g) = 1/p^l (-1)^b_0 b_l 2^{-l}$, meaning that with probability $\sim 2^{-l}$, the $l$-th bit, multiplied by $2^l$, is sent. Essentially, we assign weights from the simplex to all bits and sample them non-uniformly, but according to some prior distribution $p$.

__TopK:__ Similarly, we send the coordinate $j$ with probability $\sim |g_j|$, as follows from formula (11). It's not entirely clear why it's written so complicatedly, as it is essentially equivalent to: $p_j = |g_j| / \| g \|_1$. Again, we assign weights from the simplex to each coordinate and sample the coordinate according to $p$, resulting in a regular unbiased compressor.
Therefore we use not a uniform distribution, but $p$. According to my calculations for such a compressor $\omega = \sum 1/p_j$. And I don't understand why it is a new approach.

Maybe I'm wrong! I think it's important to explain!

---

> ### Author Rebuttal · Authors · 2025-04-01
>
> We thank the reviewer for the thoughtful and detailed feedback. Below, we address each concern and clarify the relationship between our MLMC framework and IS.
>
> **1. MLMC vs. Importance Sampling**
>
> We thank the reviewer for the insightful observation regarding the similarity between our MLMC construction and importance sampling (IS) in specific settings. That is an excellent point! We agree that in certain simple cases (bit-wise, Top-k) the MLMC estimator indeed reduces to an IS-like scheme, as you correctly pointed out. However, we respectfully argue that MLMC is not merely an instance of IS, but rather a significantly more general and natural framework for constructing unbiased estimators from biased compressors.
>
> In fact, IS can be viewed as a special case of MLMC, where sampling is performed non-uniformly over coordinates, as you have stated. MLMC provides a systematic multilevel hierarchy over increasingly accurate (less compressed) estimators, and forms unbiased estimates by applying Monte Carlo sampling over the differences between successive levels. This telescoping structure is particularly well-suited to biased compressors, where compression is naturally available at varying levels of fidelity.
>
> **Importantly, MLMC offers several advantages beyond IS:**
>
> * It can be applied immediately to any sequence of biased compressors, with no need for any manual design of coordinate-level sampling probabilities or the structure of the communicated entity, which IS requires. For e.g. top-$k$, IS uses $1/p_l \cdot g_l$ w.p. $p_l |g_l|$ to achieve the same result, and while it's straightforward in this case, it might not be as straightforward or even feasible for more complex compressors, as we elaborate below. Also, please note that this intuition regarding IS with top-$k$ was enabled by MLMC.
>
> * It is compatible with complex structured compressors that do not admit a coordinate-wise decomposition, and where IS is not naturally defined. For example, ECUQ [1] and Round-to-Nearest (RTN) [2,3] involve structured quantization (e.g., entropy constraints, grid-based rounding) for which the MLMC framework does not naturally decompose into an IS-like scheme. In such cases, it is unclear whether or how suitable IS can be defined, whereas MLMC applies seamlessly. MLMC enables these compressors to be used in a principled way to construct unbiased estimators, with automatic adaptation over compression levels.
>
> * From a practical perspective, MLMC is also more flexible and intuitive: one can simply define a sequence of biased compressors with increasing accuracy, and MLMC provides a plug-and-play mechanism for building an unbiased gradient estimate without needing to manually tune the probabilities or derive the communicated entity.
>
> We will revise the manuscript to clarify these points and include a detailed discussion comparing MLMC and IS, including when they coincide and when MLMC provides a strictly richer modeling framework. We thank you again for raising this important point.
>
> **2. Experiments**
>
> We acknowledge the reviewer’s concern regarding the ResNet18+CIFAR-10 setting. We chose this standard benchmark to align with prior works (e.g., EF21).
> Regarding accuracy, 90% requires use of Adam, LR scheduling, and more, which we did not employ as our focus was to isolate compression effects.
>
> We fully agree that larger-scale settings will better demonstrate the advantages of our method, and we ran additional NLP experiments. The results are available at "https://anonymous.4open.science/r/ICML2025MLMC-5346".
>
> **BERT Top-k**:
>
> (See repository). We evaluated our adaptive MLMC method using Top-k compression on the SST-2 benchmark using BERT finetuning. We used the AdamW optimizer.
> The folder includes 2 files showcasing the accuracy vs. \#Gbit communicated, and accuracy vs. iteration (#steps), both for M=4 machines and for k={0.01n, 0.05n, 0.1n, 0.5n}.
> We evaluated our MLMC method against EF21-SGDM, Top-k, Rand-k, and SGD (we keep this terminology, for clarity, with a slight abuse of notation, but note that the underlying optimizer for all is AdamW).
>
> As is evident by these plots, our MLMC method enjoys the fastest convergence for the same #Gbit, and it enjoys similar convergence to that of (uncompressed) SGD, and still performs better than other methods, for the same number of steps.
> These results (in addition to the ones in the paper) prove a strong advantage and efficiency of our method compared to existing methods **even for less aggressive compression**, both in communication efficiency and convergence rate, across different tasks like CV and NLP.
>
> Thank you again for your constructive feedback!
>
> [1] Dorfman et al., “DoCoFL: Downlink Compression for Cross-Device Federated Learning,” ICML 2023.
>
> [2] Gupta et al., “Quantization Robust Federated Learning for Efficient Inference on Heterogeneous Devices,” TMLR 2017.
>
> [3] Dettmers et al., “GPT3.int8(): 8-bit Matrix Multiplication for Transformers at Scale,” NeurIPS 2022.

---

> > ### Comment · Reviewer_LHZE · 2025-04-05
> >
> > > IS
> >
> > As I said in the review, I don't see much difference from the IS. The authors' response did not add anything new.There is no compressor not being IS in the paper. Are there any at all? Moreover, let us look at 220-230 (right), if there are compressors that are MLMC but not IS, then we can't do that (lines 220-230) and we have to compute 2 compression operators instead of one and we can't say nothing about efficency. Am I right?
> >
> > > Experiments on ResNet
> >
> > I ran a simple experiment with EF21 with a momentum of 0.9 in steps of 0.01 and compression of 1% and easily knocked out a accuracy of 85%
> >
> > In any case, experiments where the accuracy of the final result is 10-20% worse than what can be knocked out by simple methods without strong tuning look strange. For me it's like reporting: "all methods are bad, but ours is the best among the worst!"
> >
> > > BERT Top-k:
> >
> > Thank you! But I can't open the link. I've tried several times, I don't understand why it's like that. Are the results of these experiments the same as on ResNet? Is the quality of the training close to good results?
> >
> > None of my questions were addressed in the authors' response. Therefore, I maintain my opinion of rejection.

---

> > > ### Author Response · Authors · 2025-04-09
> > >
> > > Dear reviewer,
> > >
> > > Thank you for responding to our rebuttal, and for engaging in this discussion with us.
> > >
> > > **IS vs. MLMC**
> > >
> > > We reiterate that our MLMC method *strictly generalizes* IS, i.e. there are compressors for which IS is **not naturally defined** while our MLMC method works seamlessly. We provide the following examples:
> > >
> > > **Round-to-Nearest (RTN) compression** [1,2]: this method quantizes each element by rounding it to the nearest level on a fixed grid. The spacing of this grid is controlled by a quantization step-size. Namely, given a vector $w$, its RTN compression, $\tilde{w}$, is given by: $\tilde{w} = \delta\cdot clip(round(w/\delta),-c,c)$, where the function "round" rounds each element to its nearest integer, and the quantization step-size is given by $\delta=\frac{2c}{2^b-1}$, where the $b$ is typically $b=1,2,3,4,...$ . A *smaller* $b$ corresponds to *more aggressive* compression. **No** natural IS interpretation exists here.
> > >
> > > **Entropy-Constrained Uniform Quantization (ECUQ)** [3]: this compressor works by efficiently finding the largest number of uniformly spaced quantization levels for a given vector such that the entropy of the quantized vector (after applying entropy encoding like Huffman coding) stays within a specified bandwidth budget. **No** natural IS interpretation exists here too.
> > >
> > > Interestingly, the IS interpretation of MLMC compression seems to rise for sparsification-based compression, like top-k, bit-wise compression, but it **does not hold** for quantization-based compression, like RTN or ECUQ.
> > >
> > > We ran additional experiments on BERT fine-tuning with SST-2 comparing RTN compression with our MLMC method (with RTN-based compression). The levels of our MLMC-RTN are defined by $b$, which appears in $\delta$ and determines the quantization step-size (i.e., the extent of compression).
> > > We provide *test accuracy vs. number of steps* and *test accuracy vs. #Gbit communicated* plots for varying levels of $b$ (and hence, compression). See link below.
> > >
> > > These experiments demonstrate that our MLMC method achieves better final accuracy, faster convergence, and better communication efficiency, even though now the difference $g^l - g^{l-1}$ is not trivial (as in e.g. MLMC-top-k).
> > >
> > > Moreover, regarding the efficiency of $g^l - g^{l-1}$, during our experiments, we also calculated the average sampled MLMC-levels (which we denote in the paper by $l$ and is equivalent to $b$ in the MLMC-RTN case), and it turns out that the average level sampled is around $b\sim1.2$. **i.e, $g^l - g^{l-1}$ includes only 1-2 different numbers on average**. This makes sense since, by construction, the probability of sampling lower levels (more aggressive compression) is higher than that of higher levels, and this is consistent with classic MLMC methods. This implies that our method mostly samples lower levels (which are much cheaper to communicate) and few higher levels, but it utilizes this information very efficiently to mitigate bias and achieve superior performance across all criteria (accuracy, convergence, and communication efficiency), as our experiments show.
> > >
> > > Specifically, for additional clarity, we also provide a graph comparing our MLMC-RTN method with RTN with $b=2$. Even though the average level of MLMC-RTN is $b=1.2$ (compared to $b=2$ of regular RTN), and is thus more communication-efficient, it also achieves better accuracy and convergence.
> > >
> > > We thank you again for pointing out the connection to IS! We promise to include these new experiments and a discussion of the connection between IS and our MLMC method in the paper.
> > >
> > > **ResNet**
> > >
> > > We thank you for taking the time to run this. Our results on ResNet in our specific setting are consistent with the results obtained in previous work, see e.g. **Fig. 13 right-most plot** and **Fig.15 right-most plot** in [4] (EF21). These results are consistent with ours with similar test accuracy.
> > >
> > > In any case, our NLP experiments achieve good results (more than 90%), and this is a harder setting which demonstrates that our method works and achieves better performance across different tasks.
> > >
> > > **Link**
> > >
> > > The link works for us. It needs time to load (or maybe a different browser).
> > > In any case, we created a new anonymous repository with the results here "https://anonymous.4open.science/r/ICML2025_2-98B2/", and a dropbox with the results (anonymized account, cannot be traced back to us) here, just in case: https://www.dropbox.com/scl/fo/lmlnm9i4m51cqs185j3wh/AM6WqDl_DTLR4PI5W3dBX1I?rlkey=wr845klp30qd9ghkqmry3krhy&st=uedxj7v8&dl=0
> > >
> > > [1] Gupta et al., “Quantization Robust Federated Learning for Efficient Inference on Heterogeneous Devices,” TMLR 2017.
> > >
> > > [2] Dettmers et al., “GPT3.int8(): 8-bit Matrix Multiplication for Transformers at Scale,” NeurIPS 2022.
> > >
> > > [3] Dorfman et al., “DoCoFL: Downlink Compression for Cross-Device Federated Learning,” ICML 2023.
> > >
> > > [4] Richtárik et al. "EF21: A New, Simpler, Theoretically Better, and Practically Faster Error Feedback", NeurIPS 2021.

---

### Official Review · Reviewer_FZxr · 2025-03-14

**Overall Recommendation:** 4

**Summary:**

This paper introduces a novel Multilevel Monte Carlo (MLMC) compression scheme that leverages biased compressors to construct statistically unbiased estimates. The proposed algorithm effectively bridges the gap between biased and unbiased methods, combining the strengths of both. The empirical results show that the proposed algorithm outperforms the baselines. Theoretical analysis show that the proposed algorithm can reduce the variance incurred by the compression.

**Claims And Evidence:**

The claims made in the submission are supported by clear and convincing evidence.

**Essential References Not Discussed:**

The references look good to me.

**Experimental Designs Or Analyses:**

For the experiments, I have some concerns:

1. The experiments are very small for distributed training. I would recommend cifar-100 or even larger such as imagenet.

2. The experiments are limited to CV models. I would recommend to add some NLP (transformer) experiments.

3. Although not covered by the theoretical analysis of convergence, I would like to see some experiments of how the proposed compressor works with Adam (actually AdamW) optimizer.

4. All the experiments only show accuracy vs. #Gbit communicated. I strongly recommend to add plots of accuracy vs. steps, so that we could see the gap between the compressed methods and the optimal (final) accuracy of full-precision SGD.

**Methods And Evaluation Criteria:**

Most of the proposed methods make sense for the problem.

For the proposed methods:

I think there should be some discussion about the implementation of the proposed algorithm in the real-world distributed environment. I understand that the hardware resources are limited and the experiments in this paper seems to be simulations. However, a discussion of the implementation is still necessary. I have some concern: for Algorithm 3, the adaptive probability distribution requires the calculation of compression of all levels, hence incurring heavy computation overhead if the number of levels is large.

**Other Comments Or Suggestions:**

Please refer to the comments of the proposed method and experimental designs above.

**Other Strengths And Weaknesses:**

In overall, the idea seems very interesting and makes sense. My major concerns are about the experiments.

**Questions For Authors:**

Please refer to the comments of the proposed method and experimental designs above, and try to resolve my concerns.

**Relation To Broader Scientific Literature:**

There is nothing related to the broader scientific literature.

**Theoretical Claims:**

I've skimmed the proofs and they seems correct to me.

---

> ### Author Rebuttal · Authors · 2025-04-01
>
> We thank the reviewer for the positive evaluation of our contributions, including the novelty of our MLMC compression scheme and the theoretical analysis. We address the concerns below and will incorporate these improvements into the final version.
>
> **1. Implementation**
>
> We appreciate the reviewer’s suggestion to discuss real-world implementation. In our experiments, since hardware is limited, we used the "Multiprocessing" package to run multiple processes in parallel, each representing a different machine. This method accurately simulates a real-world parallel optimization scheme that runs on multiple machines. We will add this to the paper.
>
> **2. Computational Overhead**
>
> That is a good point. However, the adaptive sampling step in Algorithm 3 does **not** introduce significant additional computational overhead. For example, in the case of top-$k$ or $s$-top-$k$, we only need to compute each $\Delta_{t,i}^l$ (which is some norm) once per iteration, similar to what optimizers like AdaGrad already do when computing the full gradient norm. Furthermore, these norms are computed over **disjoint segments**, since $\sqrt{\Delta_{t,i}^l} = ||g_{t,i}^l - g_{t,i}^{l-1}||$, which is equivalent to the absolute value of the $l$-th largest element of $v_{t,i}$ (in top-$k$) or the norm of the segment of length $s$ with the $l$-th largest norm (in $s$-top-$k$).
> Therefore, the total computational cost is *identical* to that of computing the norm of the *full* gradient, as is done in existing adaptive methods like AdaGrad.
> A similar smart computation of the probabilities can be done for other compressors. For example, in bit-wise compressors, $g_{t,i}^l - g_{t,i}^{l-1}$ corresponds to the sign-bit and the $l$-th information bit.
>
> Moreover, the number of compression levels does not have to be linear in the dimension of the compressed entity, but could be logarithmic (which is less general, but still works). This is the "classical" case of MLMC [1] in which the quality of the "levels" (which is *inversely* correlated with the extent of compression in our case) increases exponentially and thus induces a logarithmic number of levels.
>
> Also, the computational overhead (even when calculating multiple compressions) is usually negligible compared to the overhead introduced by communication, which is the main motivation behind this work and prior works. This has been discussed extensively in prior work [2,3].
>
> **2. Experiments**
>
> We acknowledge that our current experiments are on modest-sized vision tasks, due to limited hardware. Although these are the standard benchmarks used in prior works, we agree that broader empirical validation is beneficial.
> We ran additional experiments. The results are available at "https://anonymous.4open.science/r/ICML2025MLMC-5346". We recommend downloading the repository. It includes two folders, as we elaborate below:
>
> **BERT Top-k**:
>
> We evaluated our adaptive MLMC method using Top-k compression on the SST-2 benchmark using BERT finetuning. We used the AdamW optimizer.
> The folder includes 2 files showcasing the accuracy vs. #Gbit communicated, and accuracy vs. iteration (#steps), both for M=4 machines and for k={0.01n, 0.05n, 0.1n, 0.5n}.
> We evaluated our MLMC method against EF21-SGDM, Top-k, Rand-k, and SGD (we keep this terminology, for clarity, with a slight abuse of notation, but note that the underlying optimizer for all is AdamW).
>
> As is evident by these plots, our MLMC method enjoys the fastest convergence for the same \#Gbit communicated, and it enjoys similar convergence to that of (uncompressed) SGD, and still performs better than other methods, for the same number of steps.
> These results (in addition to the ones in the paper) prove a strong advantage and efficiency of our method compared to existing methods, both in communication efficiency and convergence rate, across vastly different tasks like CV and NLP.
>
>
> **RESNET CIFAR10 Top-k**:
>
> We evaluated our adaptive MLMC method using Top-k compression on CIFAR-10 using ResNet-18 against EF21-SGDM, Top-k, Rand-k, and (uncompressed) SGD.
> The folder includes 4 files showcasing the accuracy vs. #Gbit and accuracy vs. iteration (#steps), both for M=4 and M=32 machines and for k={0.001n, 0.005n, 0.01n, 0.05n}.
>
> Our method enjoys a significant advantage over comparable methods in terms of communication efficiency, convergence speed, and final accuracy.
> Our method's advantage grows with the number of machines. In the accuracy vs. iteration plots, uncompressed SGD eventually surpasses all methods, as expected, although our method is comparable when compression is not too extreme while other compression methods still experience performance degradation.
>
> Thank you again for your constructive feedback!
>
> [1] Giles, "Multilevel Monte Carlo methods", 2013.
>
> [2] Konecny, "Federated learning: Strategies for improving communication efficiency", 2018.
>
> [3] Wang, "A field guide to federated optimization", 2021

---

### Decision · Program_Chairs · 2025-05-01

**Decision:**

Accept (poster)

**Comment:**

This paper introduces a novel Multilevel Monte Carlo (MLMC) compression scheme that leverages biased compressors to construct statistically unbiased estimates. The proposed algorithm effectively bridges the gap between biased and unbiased methods, combining the strengths of both. The empirical results show that the proposed algorithm outperforms the baselines. Theoretical analysis show that the proposed algorithm can reduce the variance incurred by the compression.

The paper received somewhat mixed reviews, with two reviewers leaning towards acceptance (scores 4 and 3), and two reviewers leaning toward rejection (scores 2 and 2). I have read the reviews, rebuttals, discussion, and skimmed through the paper, too.

However, I propose acceptance - although I believe the paper can benefit from incorporating all meaningful feedback from the reviewers, which will require a major revision. I trust the authors will do so, thoroughly. I think the idea of the paper makes sense, but it is important to spend a good amount of space to show how the new approach gives rise to new compressors that were not considered in the literature before (and not merely show that the MLMC approach can lead to compressors that are not IS).

More remarks:
- I have seen a similar multilevel approach applied to SVRG before; this paper should be cited.
- I believe a more detail comparison to EF-BV is needed since this work considers a class of compressors that interpolates between biased and unbiased ones, which is similar in spirit to what the authors aim to do here with the MLMC approach). These compressors retain some strength of the biased counterparts, yet also partially benefit from unbiasedness. Can your approach recover these compressors? Can you obtain new compressors which do not belong to the EF-BV class? I think it is important to distinguish yourself from this work (although clearly your generation method is different, I am wondering about the end result).
- Yet another important missing citation is Horvath & Richtarik, A Better Alternative to Error Feedback for Communication-Efficient Distributed Learning, arXiv:2006.11077. Just like your work, this paper tried to transform a biased compressor into an unbiased one, maintaining some benefits of the biased compressor, and yet benefiting from unbiasedness of the induced compressor. How do your compressors compare to these?
- Since you generate unbiased compressors in the end, you can use more powerful methods than EF, such as DIANA, ADIANA, Marina and Dasha. These methods rely on unbiased compressors and they are orders of magnitude better than EF21, for example, in theory, and reach the performance of biased compressors in practice (e.g., Marina with the PermK compressor does; see Szlendak et al, Permutation compressors for provably faster distributed nonconvex optimization, ICLR 2022).  How do your methods compare to, say, Marina with the PermK compressor?
- When you mention asynchronous training in the intro, it may be worthwhile mentioning the Shadowheart SGD method (Tyurin et al; Shadowheart SGD: Distributed asynchronous SGD with optimal time complexity under arbitrary computation and communication heterogeneity, NeurIPS 2024) - since this is the first parallel SGD method in terms of time complexity which caters to arbitrary computation and communication times. It utilizes unbiased compressors, and uses asynchronous communication. It was shown to be the optimal asynchronous method in terms of time complexity; and none of the methods you cite can beat it. It is relevant since your work is on the topic of communication compression.
- When mentioning local updates in the intro, you may wish to mention the ProxSkip method of Mishchenko et al (ProxSkip: Yes! Local gradient steps provably lead to communication acceleration! Finally!, ICML 2022) -- since this is the first paper showing that local updates, when performed appropriately, lead to provable communication acceleration. Interestingly, further improvement in comm complexity can be obtained by designing appropriate communication compression strategies; and this was recently shown in TAMUNA (Condat et al, TAMUNA: Doubly Accelerated Distributed Optimization with Local Training, Compression, and Partial Participation). The first work showing such double acceleration was Condat et al, Provably doubly accelerated federated learning: the first theoretically successful combination of local training and compressed communication, arXiv:2210.13277.

AC